# Multiregional transcriptomics identifies congruent consensus subtypes with prognostic value beyond tumor heterogeneity of colorectal cancer

Jonas Langerud[1,2], Ina A. Eilertsen[1], Seyed H. Moosavi[1], Solveig M. K. Klokkerud[1,2], Henrik M. Reims[3], Ingeborg F. Backe[1,4], Merete Hektoen[1], Ole H. Sjo[4], Marine Jeanmougin[1], Sabine Tejpar[5], Arild Nesbakken[2,4], Ragnhild A. Lothe[1,2] & Anita Sveen[1,2] ✉

Intra-tumor heterogeneity compromises the clinical value of transcriptomic classifications of colorectal cancer. We investigated the prognostic effect of transcriptomic heterogeneity and the potential for classifications less vulnerable to heterogeneity in a single-hospital series of 1093 tumor samples from 692 patients, including multiregional samples from 98 primary tumors and 35 primary-metastasis sets. We show that intra-tumor heterogeneity of the consensus molecular subtypes (CMS) is frequent and has poor-prognostic associations independently of tumor microenvironment markers. Multiregional transcriptomics uncover cancer cell-intrinsic and low-heterogeneity signals that recapitulate the intrinsic CMSs proposed by single-cell sequencing. Further subclassification identifies congruent CMSs that explain a larger proportion of variation in patient survival than intra-tumor heterogeneity. Plasticity is indicated by discordant intrinsic phenotypes of matched primary and metastatic tumors. We conclude that multiregional sampling reconciles the prognostic power of tumor classifications from single-cell and bulk transcriptomics in the context of intra-tumor heterogeneity, and phenotypic plasticity challenges the reconciliation of primary and metastatic subtypes.

Tumor heterogeneity is a main cause of cancer progression and treatment failure[1,2]. Most solid tumors consist of multiple subclones with different genomic profiles, metastatic potentials and responses to treatment. Colorectal cancers (CRCs) commonly have polyclonal invasion, and genomic heterogeneity of the primary tumor is associated with frequent metastasis and a poor patient survival[3,4]. However, not all tumor subclones have an impact on cancer evolution[5]. Clonal selection operates on cellular phenotypes, not genotypes, and heterogeneity appears to be decoupled at the genomic and transcriptomic levels in CRC[6]. In fact, it has been proposed that most genomic intra-tumor variation of CRCs has no major phenotypic consequences[6]. This emphasizes the importance of phenotypic plasticity[7], but the clinical relevance of intra-tumor heterogeneity is less well studied on the transcriptomic level in CRC.

[1]Department of Molecular Oncology, Institute for Cancer Research, Oslo University Hospital, Oslo, Norway. [2]Institute of Clinical Medicine, Faculty of Medicine, University of Oslo, Oslo, Norway. [3]Department of Pathology, Oslo University Hospital, Oslo, Norway. [4]Department of Gastrointestinal Surgery, Oslo University Hospital, Oslo, Norway. [5]Molecular Digestive Oncology, Department of Oncology, Katholieke Universiteit Leuven, Leuven, Belgium. ✉e-mail: anita.sveen@rr-research.no

CRC transcriptomes represent a collection of the four consensus molecular subtypes (CMS)[8]. This classification reflects tumor phenotypes and morphologies, and is associated with patient survival and drug sensitivities[9–11]. It is generally accepted that the CMS framework provides a useful starting point for further transcriptomic investigations of primary CRCs. However, the classification was developed from single bulk tissue samples of individual tumors and is vulnerable to intra-tumor heterogeneity, possibly to the point where tumors contain a mixture of all CMS classes at different proportions[10,12]. This compromises the biomarker value and the predictive power of CMS for clinical endpoints[13]. Single-cell RNA sequencing has illustrated that the diverse cell types of the tumor microenvironment contribute strongly to bulk tumor transcriptomes, as well as to intra-tumor heterogeneity and the definition of tumor subtypes[14–16]. Indeed, both the classification accuracy and the prognostic value of CMS are confounded by the tumor microenvironment[12,13,17]. Cancer cell-intrinsic expression signals are also shaped by the microenvironment[15], but might be less vulnerable to tumor heterogeneity[18]. Additional classification frameworks such as the CRC intrinsic subtypes (CRIS) and the two intrinsic CMS (iCMS) classes adhere to this rationale[19,20], but the potentially added clinical value of a cancer cell-intrinsic approach has yet to be defined. In this context, phenotypic plasticity during metastasis and metastatic heterogeneity of the classifications are likely to be relevant, as demonstrated with the original CMS[21].

Single-cell transcriptomics is a powerful technology for mapping of tumor heterogeneity. However, the high costs and technical and biological variation associated with single-cell analyses are challenges that currently limit the application to larger tumor series and the integration of datasets[22]. We hypothesized that bulk transcriptomics of multiple distinct regions of each tumor is a complementary approach in this setting, and that multiregional sampling can balance the needs to capture intra-tumor and inter-tumor variation. This has previously been used to illustrate intra-tumor heterogeneity and sampling bias in CRC, although in a limited number of tumors (up to 25)[6,23,24]. Here, we analyze multiregional and single samples of primary tumors and liver metastases ($n$ = 1093 samples from 692 patients) and show that intra-tumor heterogeneity of CMS is associated with poor survival in patients with locoregional CRC. We further show potential for transcriptomic classifications less vulnerable to tumor heterogeneity, based on cancer cell-intrinsic signals with uniform expression across tumor regions. This approach recapitulates the iCMS from single-cell sequencing, and enables further substratification into a congruent CMS framework with prognostic value in the context of intra-tumor heterogeneity. We also show plasticity of intrinsic subtypes between patient-matched primary tumors and liver metastases, and conclude that classifications of primary and metastatic CRCs are challenging to reconcile.

## Results

### Transcriptomic intra-tumor heterogeneity among multiregional samples

To get an initial overview of transcriptomic intra-tumor heterogeneity in CRC, we compared CMS classifications among 2–4 multiregional samples from each of 98 primary tumors ($n$ = 286 samples; Fig. 1a, Supplementary Data 1 and Supplementary Fig. 1). Intra-tumor CMS heterogeneity was found in 40% (seven tumors were undetermined due to unclassified samples) and reflected general transcriptomic heterogeneity, estimated as the maximum Euclidean distance of principal components (PC) 1–3 between any pair of samples per tumor (Fig. 1b). The level of heterogeneity increased with the number of samples per tumor ($p$ = 2 × 10$^{-4}$ by Kruskal–Wallis test; Supplementary Fig. 2b). Both CMS3 and CMS4 were enriched with heterogeneous tumors (Fig. 1c). CMS4 was most heterogeneous and mixed with other subtypes in 84% ($n$ = 26 of 31 tumors with at least one CMS4 sample). The most common CMS combinations were CMS2/4 ($n$ = 19, 49% of

heterogeneous tumors) and CMS1/3 ($n$ = 8, 21%), with CMS4 and CMS3 as the minor components, respectively. Combinations of CMS3/4 ($n$ = 3, 8%) and CMS1/2 ($n$ = 2, 5%) were rare. Histological cryosections of multiregional samples from three selected tumors showed morphological differences according to CMS heterogeneity (Supplementary Fig. 2c–e).

Microsatellite instability (MSI) status and $KRAS/NRAS/BRAF^{V600E}$ mutation status were concordant among all multiregional samples from each tumor with CMS heterogeneity (Fig. 1a). MSI and $BRAF^{V600E}$ mutations were strongly enriched among tumors with a major CMS1 component (MSI: 79%, odds ratio [OR] 55.6, 95% confidence interval [CI] 13.6–303.0, $p$ = 9 × 10$^{-13}$; $BRAF^{V600E}$: 82%, OR 69.0, 95% CI 16.2–394.4, $p$ = 9 × 10$^{-14}$), and not similarly frequent in CMS1-minor tumors (25% of the four tumors with <50% CMS1 samples). $KRAS/NRAS$ mutations were most frequent in CMS3 tumors (major or minor) without a CMS1 component (88% of the 16 tumors, OR 17.3, 95% CI 1.7–308.7, $p$ = 0.005).

### Transcriptomic heterogeneity primarily driven by stromal infiltration

Gene set enrichment analysis of a custom collection of gene sets relevant for CRC ($n$ = 54) showed strong enrichment with mesenchymal-like and stromal features in tumors with heterogeneous compared to homogeneous CMS classifications (Fig. 1d and Supplementary Data 2). Results were similar in subgroup analyses of each of the CMS1-3 classes separately (Supplementary Fig. 3a and Supplementary Data 3). Similar results were also found by enrichment testing of differentially expressed genes between homogeneous and heterogeneous tumors among biological processes in the Gene Ontology database (Supplementary Fig. 4 and Supplementary Data 4). Sample-wise estimates of the abundance of cancer-associated fibroblasts were higher in heterogeneous tumors, but there was no difference in the abundance of cytotoxic lymphocytes (Supplementary Fig. 5 and Supplementary Data 5). This highlighted the tumor stroma as a key component of intra-tumor transcriptomic heterogeneity, consistent with the frequent heterogeneity of CMS4.

In contrast, homogeneous tumors had strongest enrichment with signatures of cell cycle progression and regulation, as well as with MYC targets (Fig. 1d and Supplementary Fig. 3a). This was consistent with the large proportion of homogeneous tumors classified as CMS2 (48%). Notably, tumors homogeneous for CMS1 or CMS3 had no significant enrichments compared to heterogeneous tumors of the corresponding subtype, although signals were strongest for immune and metabolic processes, respectively (Supplementary Data 3). The unexpected subset of tumors homogeneous for CMS4 ($n$ = 5 microsatellite stable [MSS] tumors not exposed to treatment prior to sampling, two with $BRAF^{V600E}$ or $KRAS$ mutation) were enriched with signatures of extracellular matrix organization, the top of colonic crypts and inflammatory response (Supplementary Fig. 3a). The signature of MYC targets was depleted in homogeneous versus heterogeneous CMS4 tumors. This was due to high MYC target scores in a subset of tumors with a major CMS2 component, and indicated CMS2 admixture also in the samples classified as CMS4 (Supplementary Fig. 3b). Signatures of mesenchymal-like traits and stromal infiltration were high in CMS4 samples from both heterogeneous and homogenous tumors.

### Independent prognostic impact of intra-tumor CMS heterogeneity

Intra-tumor CMS heterogeneity was not associated with any clinicopathological parameters (Supplementary Data 5) or 5-year relapse-free survival (RFS) in the multiregional sample set ($p$ = 0.7 from Cox proportional hazards analysis; $p$ = 0.6 from corresponding analysis of general transcriptomic heterogeneity as a continuous variable). To extend the analyses to a larger patient series, we performed computational modeling of intra-tumor CMS heterogeneity in single, bulk

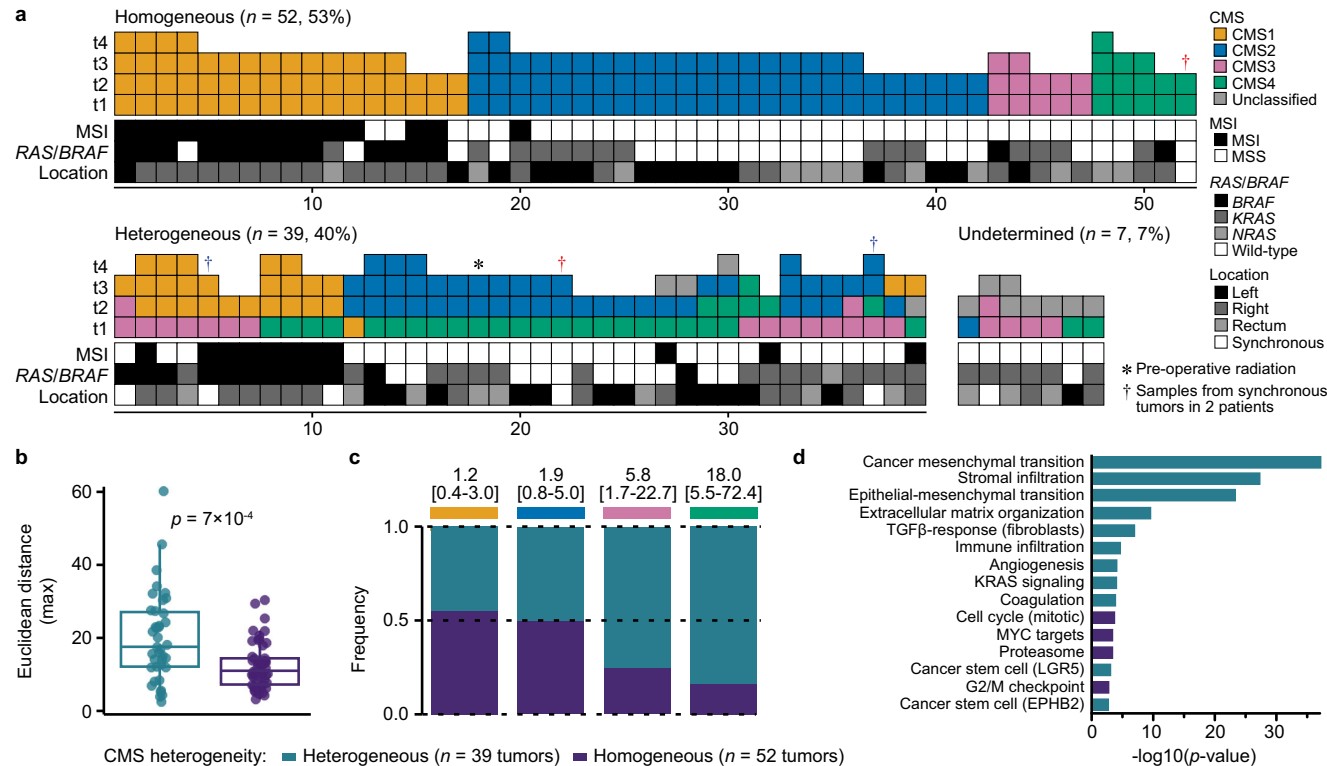

**Fig. 1 | Landscape of CMS heterogeneity among multiregional samples of primary CRCs. a** CMS classification of 2–4 multiregional samples from each of 98 primary tumors ordered according to intra-tumor classification heterogeneity. Each column represents one tumor. Spatially separated samples are annotated t1–t4 and colored according to the consensus molecular subtypes (CMS). MSI, *RAS/BRAF* and Location indicate the microsatellite instability status, mutation status (for $BRAF^{V600E}$, *KRAS* and *NRAS*) and location of each tumor in the large bowel. **b** Box plot of the general transcriptomic heterogeneity of each tumor, estimated as the maximum Euclidean distance of PC1–PC3 between any pairs of samples per tumor and plotted according to CMS heterogeneity. The center line of boxes represents the median, boxes represent the interquartile range, and whiskers represent 1.5× the interquartile range above the 75th percentile (maxima) or below the 25th percentile (minima). *p* value is two-sided and from Welch's *t*-test. Source data are provided as a Source Data file. **c** Frequency of intra-tumor heterogeneity of each CMS class. The odds ratio and 95% confidence interval for enrichment of heterogeneous tumors are showed above each class. Source data are provided as a Source Data file. **d** Bar plot of gene sets sorted according to *p* values from gene set enrichment analysis of tumors with heterogeneous versus homogeneous CMS classifications (one random sample per tumor). Enrichment in heterogeneous or homogeneous tumors is indicated by the color code. Source data are provided as a Source Data file.

tissue samples from another 418 primary CRCs (Supplementary Data 1). The approach is illustrated in Supplementary Fig. 6 and was based on enrichment scores of template gene sets of each CMS class in each sample. The template gene sets were identified from differential gene expression analysis of each CMS class versus the rest, and the enrichment scores were estimated with the R package singscore[25] (further details in "Methods"). Tumors with significant enrichments ($p < 0.05$) for more than one CMS class were considered heterogeneous, and the CMS class with the strongest enrichment was considered the major subtype. The major subtype of each tumor was largely concordant with assignments from the original random forest CMSclassifier[8], with an overall accuracy of 83% (Cohen's $\kappa = 0.72$ and 0.75 for tumors analyzed on Human Transcriptome 2.0 and Human Exon 1.0 ST arrays, respectively; Supplementary Fig. 7). The majority (88%) of discordances were due to more frequent CMS2 classifications with the enrichment analyses. CMS heterogeneity was identified in 30% of the tumors. This was less frequent than the heterogeneity observed among multiregional samples (OR 0.58, 95% CI 0.36–0.96, $p = 0.03$), and can likely be attributed to a combination of limited analytical discriminatory power (the accuracy for calling CMS heterogeneity in tumors with multiregional samples was 72%; Supplementary Fig. 8a) and the indication that heterogeneity increased with the number of samples analyzed per tumor (Supplementary Fig. 2b). The distribution of the most common CMS combinations was similar between the multiregional and single-sample tumor series, apart from a more

frequent combination of CMS3 with CMS1 in favor of CMS2 among multiregional samples (Supplementary Fig. 8b), possibly related to the enrichment with MSI tumors in this series (Supplementary Data 1).

Analysis of the combined tumor series confirmed that intra-tumor CMS heterogeneity was associated with a high abundance of cancer-associated fibroblasts, but not with any clinicopathological parameter, except for frequent CMS heterogeneity among male patients (Supplementary Data 5). Survival analysis of patients treated by complete resection of stage I–III CRC and with determined CMS heterogeneity status ($n = 387$) showed a lower 5-year RFS rate with heterogeneous (62.3%, 95% CI 54.2–71.5%) compared to homogeneous tumors (75.8%, 95% CI 70.7–81.2%; Fig. 2a). Results were similar when excluding patients with stage I tumors from the analysis (Supplementary Fig. 9a). CMS heterogeneity retained prognostic value in a multivariable Cox proportional hazards model of all clinicopathological and molecular parameters, and was the only molecular marker with a significant prognostic association (hazard ratio, HR 1.5, 95% CI 1.0–2.2, $p = 0.05$; Supplementary Data 6). Notably, CMS heterogeneity explained a larger proportion of variation in 5-year RFS (11%) than cancer-associated fibroblasts (5%; Fig. 2b and Supplementary Fig. 9b).

A stratified analysis of CMS heterogeneity according to the poor-prognostic CMS4 class (CMS4 versus CMS1-3) indicated that heterogeneous tumors with a CMS4 component (major or minor) were associated with the worst prognosis (Fig. 2c and Supplementary Fig. 9a). Heterogeneous tumors without CMS4 (different combinations

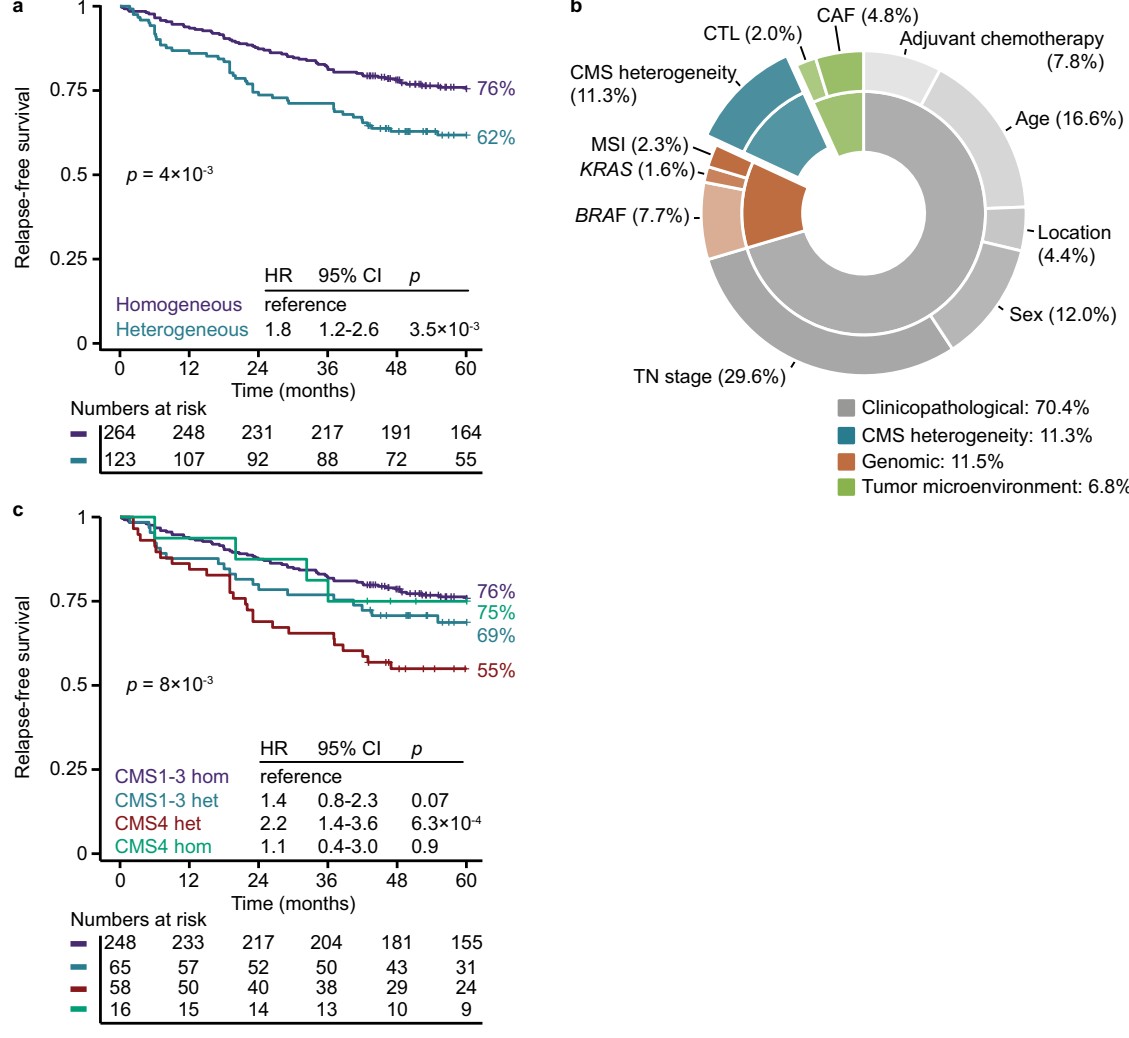

**Fig. 2 | Prognostic value of intra-tumor CMS heterogeneity. a** Relapse-free survival according to intra-tumor heterogeneity of the consensus molecular subtypes (CMS) among patients treated by complete resection of stage I–III colorectal cancer. Patients with synchronous tumors, pre-surgical radiation treatment and undetermined CMS heterogeneity were excluded from all analyses. **b** Explained variation in 5-year relapse-free survival by each variable in a multivariable Cox proportional hazards model, estimated as the percentage of the full model. **c** Survival according to intra-tumor CMS heterogeneity and stratified by CMS4 classification. Hazard ratios (HR) and 95% confidence intervals (CI) are from Cox proportional hazards analyses and p values are two-sided and from Wald tests. CAF cancer-associated fibroblasts, CTL cytotoxic lymphocytes, het heterogeneous, hom homogeneous, TN stage tumor-node stage.

of CMS1-3) had non-significant associations to worse survival relative to homogenous tumors.

## Uniform intra-tumor activity of MSI-related and oncogenic processes

The five CRIS classes derived from cancer cell-intrinsic expression signals[19] showed a similar frequency of intra-tumor heterogeneity among multiregional samples as CMS (43%, $p = 0.5$ from Fisher's exact test compared to CMS; Supplementary Data 7), although there was no significant overlap of tumors with heterogeneity according to CMS and CRIS (OR 2.0, 95% CI 0.7–5.4, $p = 0.2$; Supplementary Fig. 10). This indicated heterogeneity also within the epithelial cell compartment of CRCs and/or a stromal influence on the CRIS classification. To further investigate the basis for transcriptomic heterogeneity, we categorized protein-coding genes into three groups according to an intra-tumor heterogeneity score (ITH-score) representing intra-tumor relative to inter-tumor expression variation in the multiregional sample set (Fig. 3a, Supplementary Figs. 11 and 12, and Supplementary Data 8 and 9). The distribution of the ITH-scores was asymmetrical, with a heavy right-sided tail indicating a small subset of genes with high

intra-tumor heterogeneity (ITH-high: 5% of genes). PC1 of tumor samples from principal components analysis (PCA) based on these ITH-high genes was most strongly correlated to single-sample enrichment scores of gene sets related to stromal and mesenchymal-like features (Fig. 3b). Similar gene set results were observed for PC1 of ITH-intermediate genes (48% of genes; Supplementary Fig. 13), supporting that the majority of gene expression variation can be attributed to the stromal tumor component. In contrast, ITH-low genes (48%) were in a similar analysis associated with cancer cell-intrinsic features. PC1 of tumors based on ITH-low genes was strongly correlated to MSS/MSI-like signatures only, while PC2 was correlated to signatures of the cell cycle and proliferation (Fig. 3b). Notably, ITH-low genes showed less frequent gene set correlations with PC1 than PC2, while the opposite was observed for ITH-high and ITH-intermediate genes (OR 0.5, 95% CI 0.2–0.9, $p = 0.02$ comparing ITH-low and ITH-high genes; Fig. 3c). This suggested that genes with uniform expression across tumor regions (ITH-low) provided a more subtle tumor characterization based on the intrinsic features of cancer cells, compared to the dominating contribution from ITH-high genes and the tumor stroma.

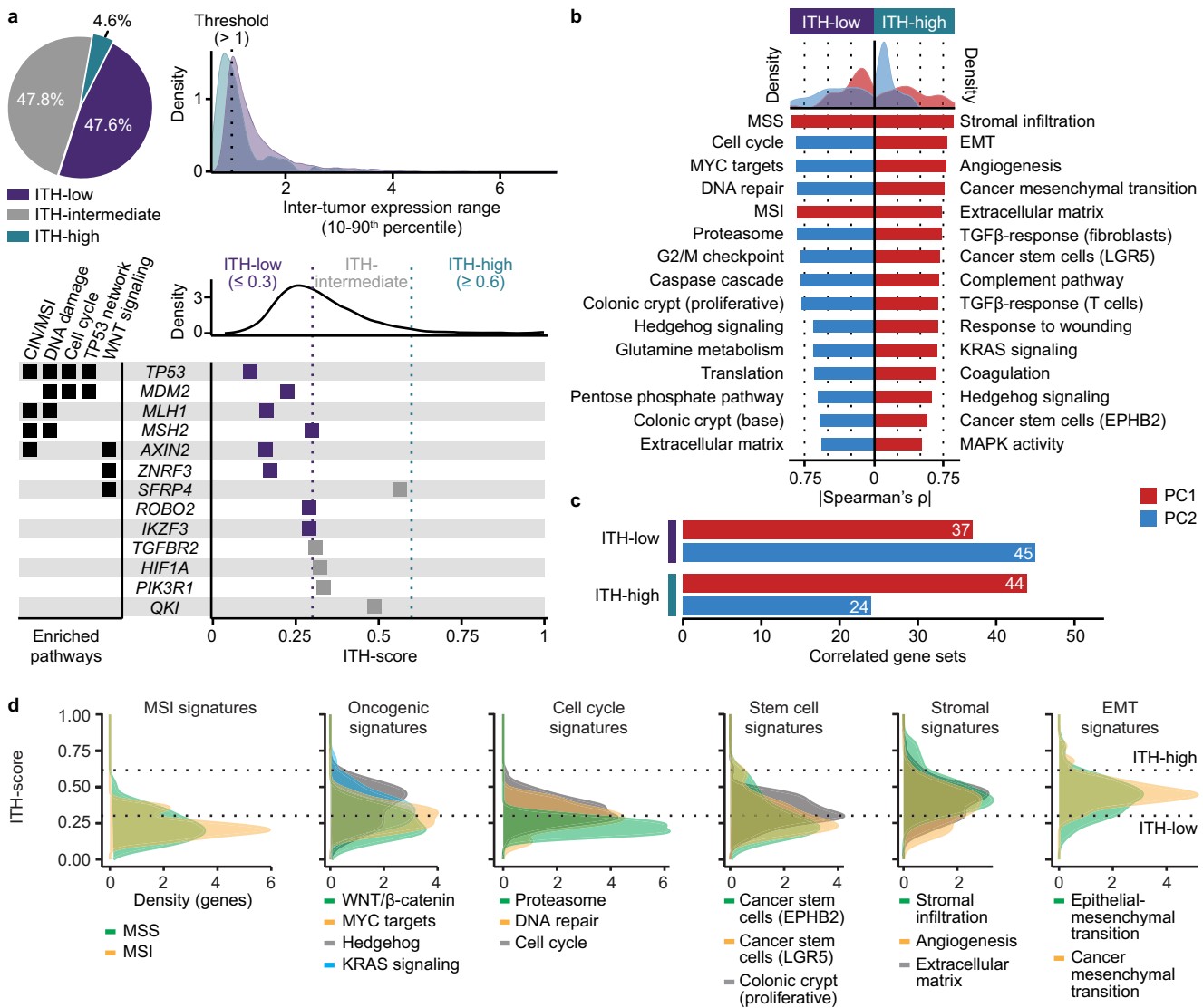

**Fig. 3 | Gene categories and enrichments according to intra-tumor heterogeneity. a** Upper part shows the proportion of genes (filtered to include only genes with inter-sample 10−90th percentile expression range >1) categorized as either ITH-low (n = 1540), ITH-intermediate (n = 1549) or ITH-high (n = 149) in the multi-regional primary tumor set (n = 286 samples). Lower part shows the density distribution of the corresponding ITH-scores, with dashed lines indicating thresholds for the three gene categories. The forest plot shows the ITH-score of genes designated as colorectal cancer-associated in the Cancer Gene Census (n = 13 genes passing the filter for the inter-sample expression range). The genes are sorted according to involvement in pathways over-represented among ITH-low genes (Wikipathways cancer; Supplementary Fig. 14). Source data are provided as a Source Data file. **b** Density plots (upper: all gene sets, n = 54) and bar plots (lower: 15 top-ranked gene sets) of Spearman's correlation coefficients (absolute values)

between PC1 or PC2 of ITH-high (right) or ITH-low (left) genes and single-sample enrichment scores of significantly correlated gene sets (p < 0.05). Source data are provided as a Source Data file. **c** Number of gene sets (of totally 54) with significant Spearman's correlations (p < 0.05) to PC1 and PC2 of ITH-high and ITH-low genes. Source data are provided as a Source Data file. **d** Distribution of ITH-scores among genes in selected signatures. The signatures were grouped into six categories, and up to three of the top-ranked gene sets according to the correlation analyses in part b are plotted per category, in addition to the WNT/β-catenin signature. Source data are provided as a Source Data file. CIN chromosomal instability, EMT epithelial-mesenchymal transition, ITH intra-tumor heterogeneity, MSI microsatellite instability, MSS microsatellite stable, PC1 and PC2 principal components 1 and 2, ρ Spearman's correlation coefficient.

The distribution of ITH-scores among scorable genes in each signature supported the results from correlation analyses, showing low ITH-scores of most MSI/MSS and cell cycle-related genes relative to epithelial-mesenchymal transition genes (Fig. 3d). Notably, genes involved in hedgehog signaling showed a near bimodal distribution of ITH-scores, and this likely accounted for the correlation of this signature with both PC1 of ITH-high genes and PC2 of ITH-low genes. Genes of WNT/β-catenin signaling and several stem cell signatures were predominantly ITH-low, but a small subset of genes in the LGR5 and EPHB2 cancer stem cell signatures had high scores, which

contributed to the correlation of these signatures with PC1 of ITH-high genes.

Cancer-critical genes, defined by the Cancer Gene Census, were underrepresented in the ITH-high category (OR 0.3, 95% CI 0.03−1.0, p = 0.05; Supplementary Data 9). ITH-low cancer-critical genes were enriched in several pathways involved in CRC tumorigenesis, such as genomic instability (chromosomal and MSI), WNT signaling and the TP53 network (Fig. 3a and Supplementary Fig. 14). ITH-high or ITH-intermediate cancer-critical genes showed no significant enrichments in a similar overrepresentation test of the Wikipathway cancer

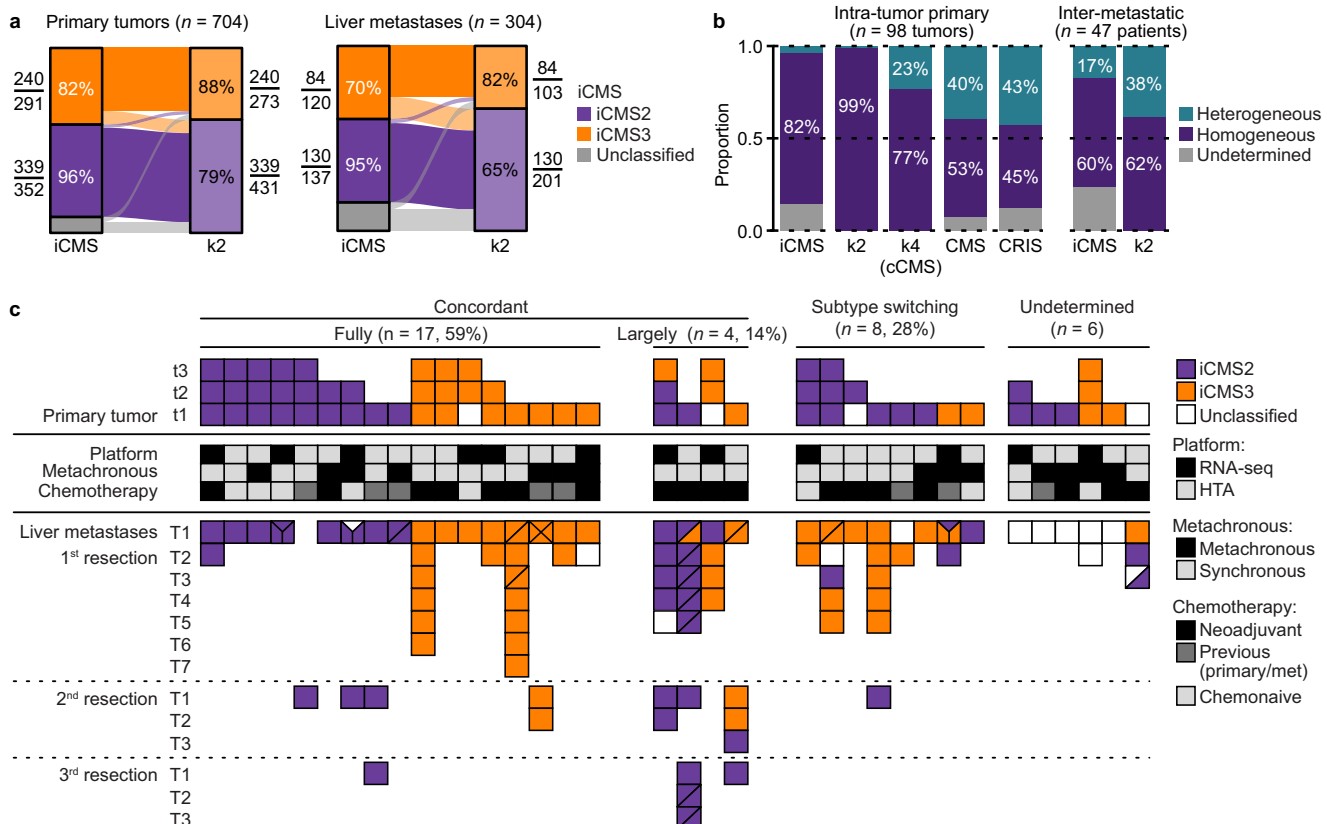

**Fig. 4 | Classification of primary tumors and liver metastases based on ITH-low and cancer cell-intrinsic genes. a** Alluvial diagrams of classification concordances between iCMS and the k2 clusters identified based on ITH-low genes among primary colorectal tumors and liver metastases. The sample overlap between classes in iCMS and k2 is indicated relative to the total number per subtype. Source data are provided as a Source Data file. **b** Proportion of primary tumors with homogeneous and heterogeneous intra-tumor classifications of multiregional samples ($n = 286$ samples) according to the indicated frameworks (k2 and k4 are based on the ITH-low genes). Proportion of patients with inter-metastatic heterogeneity among liver lesions (2–7 distinct lesions per patient, total $n = 143$ metastases) according to iCMS and the k2 clusters is plotted to the right. Source data are provided as a Source Data file. **c** iCMS classifications of matched primary tumors

and liver metastases from 35 patients ($n = 179$ samples). Each column represents one patient. For primary tumors, each square represents one multiregional sample numbered with lower case t. For liver metastases, each square represents one tumor numbered with upper case T and separated by consecutive resections, and diagonal lines indicate multiregional samples. Indicated for each patient is the platform used for gene expression analysis, diagnosis with synchronous versus metachronous metastases and exposure to chemotherapy prior to sampling. cCMS congruent consensus molecular subtypes, CRIS colorectal cancer intrinsic subtypes, iCMS intrinsic consensus molecular subtypes, ITH intra-tumor heterogeneity, k2 and k4 factorization ranks 2 and 4 from non-negative matrix factorization.

collection, suggesting that malignancy processes are not prone to intra-tumor heterogeneity on the transcriptomic level.

The ITH-scores were evaluated in a public single-cell RNA sequencing dataset of paired samples from the tumor core and border regions of six primary CRCs[15]. This confirmed that ITH-high genes had a higher expression variation among cells from paired samples than ITH-low genes ($p < 1 \times 10^{-10}$ from Welch's $t$-test; Supplementary Fig. 15).

### Evolution of ITH-low subtypes in primary-metastasis comparisons

ITH-low genes retained expression variation among tumors in the multiregional sample set, and had higher inter-tumor expression ranges (10–90th percentiles) than ITH-high (95% CI of the mean difference 0.56–0.59) or ITH-intermediate genes (95% CI 0.29–0.33; $p < 1 \times 10^{-15}$ from Welch's $t$-tests; Fig. 3a). To investigate the potential for transcriptomic classifications less prone to intra-tumor heterogeneity, we therefore performed subtype discovery by non-negative matrix factorization (NMF) of tumors based on the ITH-low genes. NMF across the full sample set ($n = 704$ samples from 516 primary tumors) at a predefined rank of $k = 2$ clusters resulted in subtypes (denoted k2) that were largely concordant with the two iCMS classes previously derived from single-cell RNA sequencing of the malignant epithelial

compartment of CRCs[20] (classification accuracy 90%, Cohen's $\kappa = 0.80$; Fig. 4a). Subtype characteristics based on gene set enrichments were also highly similar between iCMS and k2, and both frameworks were primarily distinguished by MSI/MSS-like characteristics and immune signatures (Supplementary Fig. 16). Both iCMS and k2 provided largely concordant intra-tumor classifications of multiregional primary tumor samples (82% and 99%, respectively; Fig. 4b). Collectively, this suggested that an average of three multiregional samples from each tumor could recapitulate the cancer cell-intrinsic subtypes from single-cell sequencing.

CRC liver metastases ($n = 304$ tumor samples from 179 patients) also showed concordant classifications between iCMS and the ITH-low k2 clusters (accuracy 83%, Cohen's $\kappa = 0.66$; Fig. 4a). The subtype distributions were similar among primary tumors and metastases in both frameworks (iCMS: $p = 0.8$ and k2: $p = 0.2$ from Pearson's chi-squared tests). Principal components analysis based on ITH-low genes or iCMS template genes showed no apparent distinctions according to tumor site (colorectum versus liver; Supplementary Fig. 17), supporting that also ITH-low genes primarily have cancer cell-intrinsic expression and that both classifications are directly applicable to metastatic tumors. However, the frequency of intra-patient subtype heterogeneity among metastatic lesions

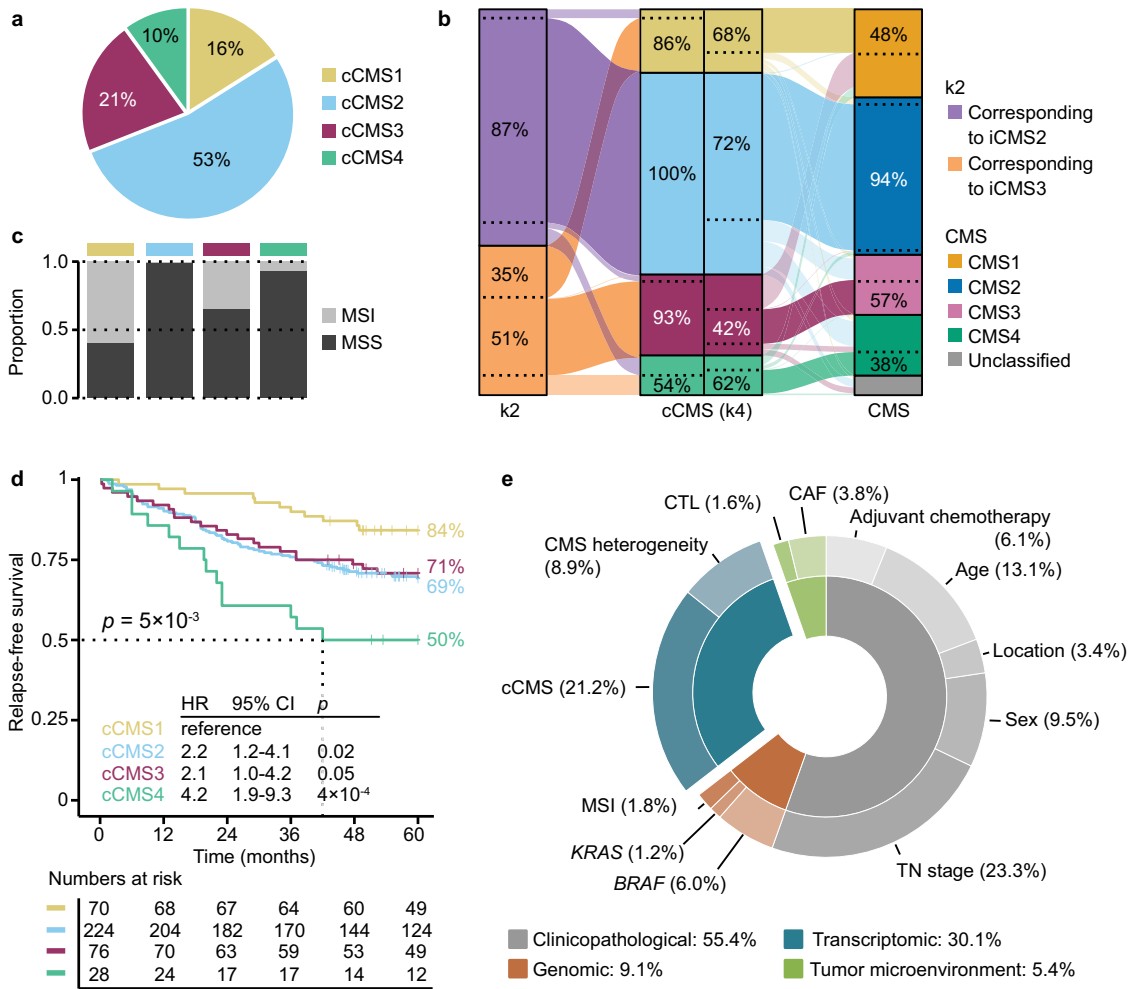

**Fig. 5 | Prognostic value of the proposed congruent CMS framework.**
**a** Proportion of the four cCMS classes among primary colorectal tumors
($n = 704$ samples from 516 tumors). Source data are provided as a Source Data file.
**b** Alluvial diagrams of classification concordances between k2 and cCMS (left;
representing k2 and k4 classifications based on ITH-low genes), as well as cCMS and
CMS (right). The sample overlap between classes is indicated relative to the total
number per subtype and illustrated by dashed lines. Source data are provided as a
Source Data file. **c** Proportion of MSI tumors across the cCMS classes. Source data
are provided as a Source Data file. **d** Relapse-free survival according to cCMS
among patients treated by complete resection of stage I–III CRC ($n = 398$). Patients
with heterogeneous intra-tumor cCMS classifications, synchronous tumors, or pre-
surgical radiation treatment were excluded from analyses. Hazard ratios (HR) and
95% confidence intervals (CI) are from Cox proportional hazards analyses and $p$
values are two-sided and from Wald tests. **e** Explained variation in 5-year relapse-
free survival ($n = 398$ patients) by each variable in a multivariable Cox proportional
hazards model, estimated as the percentage of the full model. CAF cancer-
associated fibroblasts, cCMS congruent consensus molecular subtypes, CTL cyto-
toxic lymphocytes, iCMS intrinsic consensus molecular subtypes, ITH intra-tumor
heterogeneity, k2 and k4 factorization ranks 2 and 4 from non-negative matrix
factorization, MSI microsatellite instability, MSS microsatellite stable, TN stage
tumor-node stage.

($n = 2$–7 lesions from each of 47 patients) was higher than intra-
tumor heterogeneity of the primary tumor (iCMS: $X^2 = 6.7$, $p = 0.008$
and k2: $X^2 = 35.6$, $p = 3 \times 10^{-9}$, both with one degree of freedom;
Fig. 4b). Furthermore, comparisons of patient-matched primary
tumors and liver metastases ($n = 179$ samples from 35 patients) also
showed evidence of phenotypic plasticity. Using iCMS for illustra-
tion, only 59% of evaluable patients had fully concordant classifica-
tions (17 of 29 patients, six were not evaluable due to unclassified
samples; Fig. 4c). Subtype switching between all or a majority of
primary-metastasis samples was observed in eight patients (28%).
This occurred predominantly from iCMS2 primary tumors to iCMS3
liver metastases (six of eight patients, 75%). There was no significant
association between subtype switching and use of different analysis
platforms (RNA sequencing versus Human Transcriptome 2.0 array:
$X^2 = 0.90$, $p = 0.6$), the numbers of samples/tumors per patient
($p = 0.5$ from Wilcoxon's test), diagnosis with synchronous versus
metachronous metastases ($p > 0.9$ from Pearson's chi-squared test),
or exposure to chemotherapy prior to sampling ($p = 0.9$).

## Prognostic value of congruent CMS classification based on ITH-low genes

Subtype discovery based on ITH-low genes was tested with different
sets of samples and ITH-score thresholds to evaluate a possible impact
on classification results (Supplementary Figs. 18–21; details in "Meth-
ods"). NMF at $k = 4$ or $k = 5$ were identified as the best sample cluster-
ings, but k5 included two clusters with similar characteristics, and k4
was therefore used for further analyses of the complete primary tumor
series. This ITH-low classification approach indicated potential for an
intrinsic classification with a higher resolution than the two-state iCMS
framework. The k4 clusters ranged in size from 10% to 53% of samples
and subdivided each of the k2 clusters, most prominently the cluster
corresponding to iCMS3 (Fig. 5a, b; the iCMS framework was similarly
split, Supplementary Fig. 22). The iCMS3-corresponding cluster was
split into one cluster with strong immune signals and one with high
expression of genes encoding extracellular matrix remodeling pro-
teins (*FN1* and *SPP1*[26]), while the largest and remaining cluster had high
relative expression of genes involved in maintenance of the secretory

intestinal stem cell niche (for example, *REG4*, *TFF1*, *FCGBP* and *AGR2*[27–30]; analyzing ITH-low genes only; Supplementary Fig. 23).

The k4 clusters were not independent of the original CMS ($X^2$ = 589, nine degrees of freedom, $p < 3 \times 10^{-16}$), and 67% of samples showed concordant classifications with CMS (classification accuracy 68%, Cohen's $\kappa$ = 0.52; Fig. 5b). The strongest discordance was found for the cluster corresponding to the original CMS3, and this cluster was split between CMS1 and CMS3. Samples with discordant classifications were located near the class boundaries in PCA (Supplementary Fig. 24). Gene set enrichment analyses further demonstrated that each of the four sample clusters defined by ITH-low genes had similar characteristics to the corresponding CMS class (Supplementary Fig. 25), and the k4 clusters were therefore termed congruent CMS (cCMS). The largest difference was enrichment with several signatures in cCMS1 and cCMS2 that in the original CMS classification were characteristic of CMS2 only, such as MYC targets and cell cycle signatures. This can likely be attributed to heterogeneity of the subtypes, tumors, or samples, and cCMS1 and cCMS2 samples that were not of the corresponding original CMS class were more frequently from tumors with CMS heterogeneity (Supplementary Fig. 25). Intra-tumor classification concordances of multiregional samples were higher for cCMS (77% of tumors) than for the original CMS (53%, OR 2.4, 95% CI 1.3–4.8) and CRIS frameworks (45%, OR 3.1, 95% CI 1.6–6.2; Fig. 4b), indicating stronger robustness to intra-tumor heterogeneity.

Genomic markers (MSI and *BRAF*^*V600E*) and tumor microenvironment markers (cancer-associated fibroblasts and cytotoxic lymphocytes) showed similar subtype associations in the cCMS and original CMS frameworks, with the exception that *KRAS* mutations were not skewed among cCMS classes (Supplementary Data 10). Consistent with the strong enrichment for MSI-like characteristics among ITH-low genes, MSI status was strongly skewed according to cCMS ($X^2$ = 174, three degrees of freedom, $p < 3 \times 10^{-16}$) and enriched in both cCMS1 and cCMS3 (OR 14.7, $p < 3 \times 10^{-16}$ and OR 2.9, $p = 4 \times 10^{-5}$, respectively; Fig. 5c). However, repeated subtype discovery of MSS tumors only (based on ITH-low genes; Supplementary Fig. 18) largely recapitulated the cCMS classification (accuracy 90%, Cohen's $\kappa$ = 0.82; Supplementary Fig. 26), indicating that the transcriptomic MSI-like features of the ITH-low genes extended beyond the genomic phenotype.

Clinicopathological associations were also similar between the cCMS and original CMS frameworks, although patient age at diagnosis was skewed according to cCMS (older age with cCMS1 and younger with cCMS4: OR 4.5, $p = 1 \times 10^{-4}$; Supplementary Data 10). Survival analyses of patients with concordant intra-tumor classifications (no subtyping heterogeneity among multiregional samples) showed that cCMS had strong associations to 5-year RFS in stage I–III CRC ($n$ = 398 patients; Fig. 5d). Higher and lower RFS rates were observed with cCMS1 and cCMS4 tumors, respectively, relative to each of the other subtypes. These prognostic associations were consistent with cCMS1 consisting primarily of an immune-active subset of iCMS3 tumors, and cCMS4 of iCMS3 tumors (but also a proportion of iCMS2) with active extracellular matrix remodeling, which can promote immune suppression and metastasis[15,26,31] (Supplementary Figs. 22 and 23). Results were similar with 5-year overall survival as the endpoint, and when excluding patients with stage I tumors (Supplementary Fig. 27). The cCMS framework retained prognostic value when added to the multivariable survival model shown in Fig. 2b (Table 1 and Supplementary Data 11), and explained a larger proportion of variation in 5-year RFS (21%) than CMS heterogeneity (9%) and any other molecular variable (Fig. 5e). Notably, the original CMS classes had no prognostic value in this subset of patients (Supplementary Fig. 28).

### ITH-low classifications of external primary tumor series
Subtype discovery based on the ITH-low genes was also performed in two external datasets for validation purposes (Supplementary Data 12).

As in the in-house series, NMF clustering of tumors in GSE39582 ($n$ = 566)[32] at a predefined rank of $k$ = 2 was concordant with iCMS classification (accuracy 89%, Cohen's $\kappa$ = 0.77; Fig. 6a). NMF at $k$ = 4 failed to distinguish tumors with immune and stromal infiltration (Supplementary Fig. 29a, b), but clustering at $k$ = 5 identified subtypes with highly similar characteristics to the k5 clusters in the in-house series (Fig. 6b and Supplementary Fig. 19b, c). The two clusters from k5 that corresponded to the original CMS2 showed no clear distinctions in the custom gene set collection for either tumor series (Fig. 6d and Supplementary Fig. 20). However, the consistency of the two clusters in both tumor series supported a potential for subclassification of the large CMS2 group, and pathway enrichment testing of differentially expressed genes between the clusters in the KEGG pathway database indicated separation based on signatures of bacterial and viral infection, the cell cycle, and several oncogenic or tumor suppressor signaling pathways (Supplementary Fig. 29c). The k5 clusters also had prognostic associations among stage I–III cancers in the GSE39582 series and identified a subset of mesenchymal-like tumors associated with a low 5-year RFS rate (Fig. 6e). This subtype (denoted NMF4) had only partial overlap with the original CMS4 (Fig. 6b) and improved the prognostic stratification of tumors relative to the original CMS classification (Supplementary Fig. 29d). The CMS1-corresponding cluster (denoted NMF1) had a higher 5-year RFS rate than the other subtypes among stage III cancers, but not among stage II (Supplementary Fig. 29e).

Clustering of tumors in The Cancer Genome Atlas series (TCGA; $n$ = 573)[33] based on ITH-low genes at $k$ = 2 segregated a small subset of exclusively MSS tumors (9%), and showed little correspondence with iCMS (Supplementary Fig. 30a). Clustering at $k$ = 5 failed to distinguish tumors with immune and stromal infiltration (Supplementary Fig. 30b), similarly to the k4 clusters in GSE39582, and a higher factorization rank was therefore used. Clustering at $k$ = 6 identified subtypes that showed a similar degree of overlap with the original CMS classification (accuracy 58% and Cohen's $\kappa$ = 0.41) as the k5 clusters in both the in-house and GSE39582 series (in-house: accuracy 63% and Cohen's $\kappa$ = 0.44; GSE39582: accuracy 67% and Cohen's $\kappa$ = 0.52; Fig. 6b, c and Supplementary Fig. 19c). The k6 clusters in TCGA included two CMS2-corresponding clusters (termed NMF2 and NMF2.5) and two CMS3-corresponding clusters (termed NMF3 and NMF3.5; Fig. 6d). Notably, merging of the two CMS2-corresponding clusters (NMF2 and NMF2.5) versus the rest (NMF1, NMF3, NMF3.5, NMF4) provided a two-state classification concordant with iCMS (accuracy 85% and Cohen's $\kappa$ = 0.69; Fig. 6b). Furthermore, comparisons of the two CMS2-corresponding clusters by pathway enrichment analyses of differentially expressed genes showed several of the same distinctions in TCGA as in the in-house and GSE39582 series, providing further support for the subclassification of CMS2 based on characteristics such as bacterial or viral infections (Fig. 6f). The two CMS3-corresponding clusters in TCGA (NMF3 and NMF3.5) were primarily distinguished based on MSI status and signatures of the bottom versus top of colonic crypts (Fig. 6c, d). Collectively, these validation analyses suggested that the ITH-low genes distinguished tumors in independent series according to the same biological and clinicopathological characteristics, although with a varying number of sample clusters (factorization ranks).

## Discussion
Multiregional tumor transcriptomics represents a feasible approach to balance the needs to capture both intra-tumor and inter-tumor gene expression variation. We analyzed a large series of multiregional samples from primary CRCs and used this to distinguish heterogeneous and uniform expression features across tumor regions, while retaining information of tumor subtypes (that is, variation across tumors). Three bulk samples per tumor could recapitulate cancer cell-intrinsic expression patterns and subtypes that were less vulnerable to

**Table 1 | Multivariable survival analysis of clinicopathological and molecular features in patients with stage I-III CRC**

| Variable | | Patients[a] | Five-year relapse-free survival[b] | | |
|---|---|---|---|---|---|
| | | n (%) | HR [95% CI] | p-value | cox.zph p-value |
| Total | | 398 (100) | | | 0.2 |
| Sex | Female | 204 (51) | Reference | | 0.2 |
| | Male | 194 (49) | 1.5 [1.0-2.2] | 0.05 | |
| Age (continuous) | | 398 (100) | 1.0 [1.0-1.1] | **0.0007** | **0.009** |
| Tumor location | Right | 176 (44) | Reference | | 1.0 |
| | Left | 125 (31) | 0.9 [0.6-1.5] | 0.7 | |
| | Rectum | 97 (24) | 1.0 [0.6-1.7] | 0.9 | |
| TNM stage | I | 95 (24) | Reference | | 0.4 |
| | II | 176 (44) | 1.2 [0.7-2.2] | 0.5 | |
| | III | 127 (32) | 2.1 [1.1-3.9] | **0.02** | |
| Adjuvant chemotherapy | No | 327 (82) | Reference | | 0.4 |
| | Yes | 64 (16) | 1.2 [0.6-2.2] | 0.7 | |
| | Unknown | 7 (2) | 2.4 [0.7-8.1] | 0.1 | |
| MSI status | MSS | 320 (80) | Reference | | 0.6 |
| | MSI | 78 (20) | 0.8 [0.3-1.7] | 0.5 | |
| KRAS status | Wild-type | 258 (65) | Reference | | 0.06 |
| | Mutation | 140 (35) | 1.1 [0.7-1.8] | 0.6 | |
| BRAF[V600E] status | Wild-type | 328 (82) | Reference | | 0.4 |
| | Mutation | 70 (18) | 2.6 [1.3-5.3] | **0.01** | |
| CTL-score (continuous) | | 398 (100) | 1.2 [0.5-3.2] | 0.7 | 0.6 |
| CAF-score (continuous) | | 398 (100) | 1.1 [0.8-1.5] | 0.7 | 1.0 |
| CMS heterogeneity | Homogeneous | 257 (65) | Reference | | 0.5 |
| | Heterogeneous | 115 (29) | 1.6 [1.1-2.4] | **0.03** | |
| | Undetermined | 26 (7) | 2.2 [1.0-4.7] | **0.05** | |
| cCMS | cCMS1 | 70 (18) | Reference | | 0.5 |
| | cCMS2 | 224 (55) | 3.7 [1.6-8.7] | **0.002** | |
| | cCMS3 | 76 (19) | 2.3 [1.0-5.2] | **0.04** | |
| | cCMS4 | 28 (7) | 4.3 [1.6-11.3] | **0.003** | |

*CAF* cancer-associated fibroblasts, *CMS* consensus molecular subtypes, *cCMS* congruent consensus molecular subtypes, *CTL* cytotoxic lymphocytes, *MSI* microsatellite instability, *MSS* microsatellite stable, *TNM* tumor-node-metastasis.
[a]Excluding patients with synchronous tumors, heterogeneous cCMS classification, pre-surgical chemoradiation and residual tumor status 1 or 2.
[b]Hazard ratios (HR) and 95% confidence intervals (CI) are from a multivariable Cox proportional hazards model, *p* values are two-sided and from Wald tests, and cox.zph *p* values are from tests of the proportional hazards assumption. Statistically significant *p* values are highlighted in bold. Results were similar in a stratified analysis according to the variable breaking the proportional hazards assumption (patient age; Supplementary Data 11).

intra-tumor heterogeneity. While single-cell RNA sequencing was needed to initially delineate these patterns and define the iCMS classification[20], this study showed potential to expand on the knowledge and suggested a further substratification of ITH-low intrinsic subtypes. This resulted in a split predominantly of the subtype corresponding to iCMS3. However, the split was not primarily defined by MSI status, as proposed with the refined IMF (intrinsic-microsatellite-fibrosis) classification[20]. The ITH-low subtypes rather converged on having the same discriminatory biological features as the original CMS[8], although it has previously been shown that the original CMS classifier is depleted of genes with uniform expression among tumor glands[6]. Nonetheless, the convergence is consistent with the assumption that the tumor microenvironment is at least partly shaped by malignant epithelial cells and that the tumor epithelium can recapitulate the CMS classification[16]. This was also the premise for the successful classification of diverse pre-clinical model systems according to CMS[34,35]. Overall, this supports that CMS-related features provide a bona fide phenotypic stratification of CRCs, but the precise cellular interactions defining the subtypes with a rich microenvironment component are still to be uncovered. Spatial transcriptomics has potential to delineate such interactions, as recently shown with the detailed description of the interaction networks of immune and malignant cells according to MSI status of the tumors[36]. In this context,

the congruent subtypes proposed in this study can be considered as an alternative CMS classification that is based on cancer cell-intrinsic template genes and therefore less vulnerable to intra-tumor heterogeneity. However, this interpretation does not fully account for the stronger prognostic power of the congruent subtypes.

In contrast to the original CMS classification, the congruent CMS classes provided substantial prognostic value beyond both intra-tumor heterogeneity and the tumor microenvironment components in patients with locoregional cancer. However, the two prognostic subtypes (cCMS1 and cCMS4) constituted only one-fourth of the tumors in total. Both prognostic subtypes were dominated by tumors corresponding to iCMS3, but included only approximately half of all iCMS3 tumors. This is largely consistent with the original publication showing that the binary iCMS classification is not prognostic[20]. A poor patient survival was found with fibrotic iCMS3 tumors only, and this subtype constituted ~30% of iCMS3 tumors and 14% in total. Notably, the proposed cCMS classification additionally identified a subset of mostly iCMS3 tumors with a favorable prognostic association, independently of MSI status. This further supports that substratification of iCMS is needed in the evaluation of patient prognosis, and the proposed cCMS might reconcile the single-cell-derived iCMS and the original bulk transcriptomics-derived CMS for this purpose. Application of cCMS to additional tumors is not dependent on multiregional sampling and can

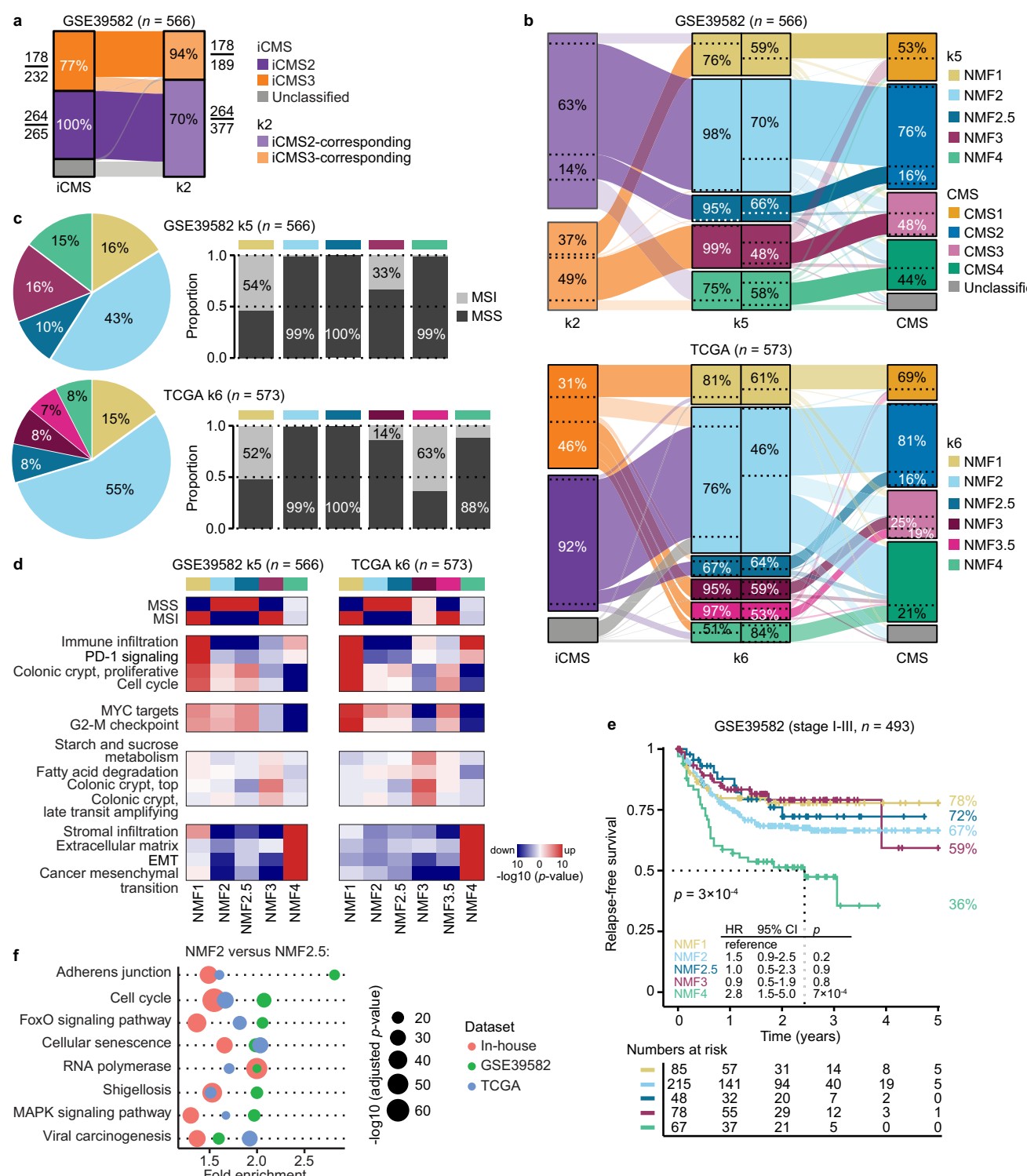

**a** GSE39582 (*n* = 566)

**b** GSE39582 (*n* = 566)

**c** GSE39582 k5 (*n* = 566)

TCGA k6 (*n* = 573)

**d** GSE39582 k5 (*n* = 566)  TCGA k6 (*n* = 573)

**e** GSE39582 (stage I-III, *n* = 493)

|  | HR | 95% CI | *p* |
|---|---|---|---|
| NMF1 | reference | | |
| NMF2 | 1.5 | 0.9-2.5 | 0.2 |
| NMF2.5 | 1.0 | 0.5-2.3 | 0.9 |
| NMF3 | 0.9 | 0.5-1.9 | 0.8 |
| NMF4 | 2.8 | 1.5-5.0 | 7×10⁻⁴ |

Numbers at risk

| 85 | 57 | 31 | 14 | 8 | 5 |
| 215 | 141 | 94 | 40 | 19 | 5 |
| 48 | 32 | 20 | 7 | 2 | 0 |
| 78 | 55 | 29 | 12 | 3 | 1 |
| 67 | 37 | 21 | 5 | 0 | 0 |

**f** NMF2 versus NMF2.5:

be done based on the ITH-low genes, as illustrated in two external primary tumor series. However, different factorization ranks were needed to identify corresponding subtypes in the different series, and the optimal number of ITH-low subtypes remains inconclusive. It is not clear whether this inconsistency is related to technical variation from use of different gene expression platforms or to biological differences among the series. Nonetheless, subtypes with similar gene expression characteristics to the four cCMS classes were found in both external series, and a potential for subclassification of the large and heterogeneous group of CMS2-corresponding tumors based on characteristics such as bacterial or viral infections was supported in all the series

analyzed. This subclassification is also consistent with a microbiome-dependent subtype proposed in a previous study[37]. Additional translational studies are needed to consolidate the ITH-low classification, support the prognostic value and explore additional clinical relevance, for example, by associations with drug sensitivities.

Our work also showed prognostic relevance of intra-tumor heterogeneity. These results are highly similar to a previous study based on computational deconvolution of intra-tumor CMS heterogeneity in single samples of stage III colon cancers[38], and supported a poor prognosis with a minor CMS4 component in particular. Notably, the CMS combinations frequently observed by multiregional sampling, or

**Fig. 6 | Classification of external primary tumor series based on ITH-low genes.** Alluvial diagrams of classification concordances (**a**) between iCMS and the ITH-low k2 clusters among tumors in the GSE39582 series, (**b**) the ITH-low k2 and k5 clusters and the original CMS in GSE39582 (top), as well as the iCMS, ITH-low k6 and original CMS in TCGA (bottom). The sample overlap is indicated relative to the total number per subtype. Source data are provided as a Source Data file. **c** Pie charts of the proportion of tumors in each of the ITH-low k5 and k6 clusters in GSE39582 and TCGA, respectively, bar charts of the proportion of MSI tumors in each cluster, and (**d**) heat maps of *p* values from gene set enrichment analyses (log10-scale; red: positive; blue: negative). Gene sets were from the custom collection and selected to include the same as in the corresponding analysis of the in-house tumor series (Supplementary Fig. 20). Source data are provided as a Source Data file. **e** Five-year relapse-free survival according to the ITH-low k5 clusters in patients with stage I–III CRC in GSE39582. Hazard ratios (HR) and 95% confidence intervals (CI) are from

Cox proportional hazards analyses and *p* values are two-sided and from Wald tests. The two-sided log-rank *p* value across subtypes is also given. **f** Common pathway enrichments in the KEGG database of differentially expressed genes between the two CMS2-corresponding subtypes (NMF2 and NMF2.5) from ITH-low k5 clustering of the in-house and GSE39582 series, and ITH-low k6 clustering of the TCGA series. The eight pathways with highest max enrichment score among the common pathways of the top-50 most significantly enriched in each tumor series were selected for plotting. Dot sizes indicate the significance level (Benjamini–Hochberg adjusted *p* value on log10-scale). Source data are provided as a Source Data file. cCMS congruent consensus molecular subtypes, EMT epithelial-mesenchymal transition, iCMS intrinsic consensus molecular subtypes, ITH intra-tumor heterogeneity, k2 and k4 factorization ranks 2 and 4 from NMF, MSI microsatellite instability, MSS microsatellite stable, NMF non-negative matrix factorization, TCGA The Cancer Genome Atlas.

estimated by computational enrichments, were similar to results from single-cell sequencing of a smaller tumor series[15]. Nonetheless, tumors analyzed by the largest number of multiregional samples were frequently scored as heterogeneous, and it is likely that CMS heterogeneity is underestimated in studies based on bulk transcriptomics. Even small tumor subclones can have clinical relevance with respect to development of resistance during treatment[39], but we cannot conclude on the lower limit of what can be considered prognostically relevant transcriptomic heterogeneity, or on the number of samples needed to detect this. According to the "big bang" model of CRC development, invasive cancers have spatially intermixed subclones on the genomic level[40]. This supports the potential to capture heterogeneity with a small number of samples, although a potential caveat is that such clonal intermixing is not necessarily reflected on the transcriptomic and phenotypic levels.

Transcriptomic subtypes based on cancer cell-intrinsic signals have the presumed advantage of being applicable to both primary and metastatic tumors, without the need to adapt the classification approach. This was supported by PCA based on the ITH-low genes detected in this study and on the iCMS template genes, both showing intermingling of primary tumors and liver metastasis, which is in contrast to results based on unselected genes[41]. In further support of the appropriateness of intrinsic classifications for metastatic tumors, we did not observe any subtype depletions or shift in the distribution of iCMS classes between primary tumors and liver metastases. This was unexpected based on the strong depletion of the original CMS1 and CMS3 classes among metastases[21], which would suggest a depletion also of iCMS3. Nonetheless, subtype switching of iCMS between matched primary and metastatic tumors was observed in almost a third of patients. This was noteworthy in particular since iCMS is only a two-state classification. Switches of cancer cell-intrinsic classes can be due to either clonal evolution or transition of differentiation states. Clonal evolution and selection of the minor clone is a possible explanation based on the non-exclusivity of iCMS classes among cells in each primary tumor[20]. However, phenotypic plasticity and cellular differentiation and dedifferentiation might be an essential trait for cancer metastasis[42]. The dynamic cellular states observed in models of CRC metastasis[43] open up the possibility for cells to even transition between iCMS classes during metastasis and to eventually end up in their original iCMS in established metastatic tumors. According to this view, the heterogeneity is dependent on the timing of sampling and would therefore be underestimated in our study. The most frequently observed switch from iCMS2 primary tumors to iCMS3 liver metastases is consistent with dedifferentiation from an LGR5-positive stem cell[20], although our study was not sufficiently powered to confirm this predilection. Nonetheless, the profound phenotypic plasticity observed in at least a subset of patients challenges the potential reconciliation of subtyping schemes of primary and metastatic tumors, also of the congruent CMS proposed here. This supports the need for a de novo classification of metastases based on their in situ cellular states[41].

In conclusion, we describe transcriptomic features with prognostic value independently of the tumor microenvironment and in the context of intra-tumor heterogeneity of CRC. Multiregional transcriptomics captured cancer cell-intrinsic features with low intra-tumor heterogeneity, and identified congruent CMS classes that appeared to reconcile the prognostic potential of current classifications derived from single-cell and bulk transcriptomics. However, evidence of phenotypic plasticity during metastasis, even with a two-state cancer cell-intrinsic classification, indicated that reconciliation of primary and metastatic subtyping frameworks is challenging.

## Methods

### Patient material

The study has been approved by the Regional Committee for Medical and Health Research Ethics, South Eastern Norway (REC numbers 1.2005.1629 and 2010/1805). All patients provided written informed consent, and the study was conducted in accordance with the Declaration of Helsinki. All patients were treated according to national standard guidelines. Patient sex was assigned as registered in the medical records at Oslo University Hospital, Norway, and was not considered in the study design.

A total of 1093 fresh frozen primary tumor and liver metastasis samples from 692 patients treated surgically for primary and/or metastatic CRC at Oslo University Hospital were analyzed for gene expression in the study. Samples were taken from surgical specimens at the operating theater and prior to pathological examination. Two to four multiregional samples (mean of 2.9) were taken from spatially distinct areas of each of 98 primary tumors from 96 patients treated in 2015 and 2016 (*n* = 286 samples; Supplementary Data 1). Tumors with multiregional sampling had a diameter of at least 15 mm (median diameter of 40 mm, 95% CI 35–40), and multiple samples were not taken unless clearly spatially separated. There was no association between tumor size and the number of sampled regions from each tumor (*p* = 0.4 by Kruskal–Wallis test; Supplementary Fig. 2a). RNA and DNA were extracted using the Qiagen AllPrep DNA/RNA/miRNA Universal Kit or DNA/RNA Mini Kit in accordance with the manufacturer's protocol (Qiagen GmbH, Hilden, Germany). Cryosections of selected samples were stained with hematoxylin and eosin and evaluated for histologic tumor grade according to the WHO classification (5th edition)[44], as well as morphological patterns previously associated with an image-based CMS classification[10].

Molecular data from single primary tumor samples of an additional 418 patients treated between 2005 and 2013 have previously been published (Supplementary Data 1)[9]. Liver metastasis samples (*n* = 338) were collected from 191 patients treated by hepatic resection between 2013 and 2018, and molecular data have previously been published for the majority (*n* = 280 samples from 1–7 liver lesions of each of 171 patients)[41]. Patient-matched sets of primary tumor and metastasis samples were available from 35 patients (total *n* = 179 samples). The primary tumor from 22 of these patients

($n = 51$ samples) were included for longitudinal comparisons only and not otherwise analyzed in the study. Twenty-one (60%) of the patients with primary-metastasis samples had synchronous metastatic disease (liver metastases diagnosed within 6 months of the primary tumor), and 14 (40%) had metachronous metastases. Eighteen (51%) received neoadjuvant chemotherapy for the sampled metastases, eight (23%) had previously received chemotherapy for primary and/or metastatic CRC, and nine (26%) were chemonaive at the time of sampling.

Processed gene expression data and metadata of two external primary tumor series were downloaded from the SAGE Bionetworks Synapse platform (https://www.synapse.org/#!Synapse:syn2634724) and used for validation analyses. This included 566 tumors from the GSE39582 series and 573 tumors from TCGA (Supplementary Data 12)[32,33]. Processed single-cell RNA sequencing data and metadata for totally 17,678 cells from 12 paired samples of the tumor core and tumor border regions of each of six primary CRCs were downloaded from NCBI's Gene Expression Omnibus (GEO) with accession number GSE144735[15].

## MSI and mutation analyses

MSI status of the multiregional primary tumor series was determined by PCR-based analyses of mononucleotide repeat markers using the Promega MSI Analysis System in accordance with the manufacturer's protocol (Promega, Madison, WI, USA). Mutational hotspots in *KRAS* and *NRAS* exons 2–4, as well as *BRAF* exon 15 (including codon 600) were analyzed by Sanger sequencing using the Cycle Sequencing Kit and 3730 DNA Analyzer (Applied Biosystems, Waltham, MA, USA) as previously described[45]. One randomly selected sample per tumor and all samples from tumors with discordant CMS classifications were analyzed ($n = 158$ samples).

Tumor content has been confirmed in the multiregional samples by deep sequencing of a custom panel of twenty genes, using matched normal colonic mucosa samples as reference. Homogenous somatic single nucleotide variants or short insertion or deletions in *APC*, *TP53*, *KRAS*, *NRAS*, *BRAF*, *PIK3CA* and/or *FBXW7* (same mutation present in all samples per tumor) were found in all tumors except one that was not scored, all with a mutant allele fraction above 5% (the data and additional details will be published elsewhere).

## Gene expression profiling and data processing

All in-house tumor samples have been analyzed for gene expression on high-resolution platforms ($n = 1093$; Supplementary Fig. 1). Multiregional primary tumor samples were analyzed on Affymetrix Human Transcriptome 2.0 arrays (HTA), using 100 ng of total RNA as input and following the manufacturer's protocol (Thermo Fisher Scientific, Waltham, MA, USA). The extended single-sample primary tumor set has previously been analyzed on HTA ($n = 217$) or Affymetrix GeneChip Human Exon 1.0 ST arrays (HuEx; $n = 201$)[9]. Patient-matched primary-metastasis samples were analyzed on HTA ($n = 23$ patients and 116 samples) or by total RNA sequencing ($n = 12$ patients and 63 samples). The remaining liver metastases samples have been analyzed on HTA arrays[41]. RNA sequencing was performed in $2 \times 101$ base-pair paired-end mode on the Illumina HiSeq 4000 platform (Illumina, San Diego, CA, USA) at the Oslo University Hospital Genomics Core Facility to a median depth of $52.6 \times 10^6$ uniquely mapped read pairs per sample ($10$–$90$th percentile $40.5 \times 10^6$–$71.6 \times 10^6$). Sample preparation was performed with ribosomal RNA depletion using the Ribo-Zero Gold rRNA removal kit and sequence library generation with the TruSeq Stranded Total RNA Library Prep Gold kit (Illumina).

Raw intensity data CEL-files from microarray experiments were processed in five separate datasets (multiregional primary tumor samples, primary single-sample HTA, primary single-sample HuEx, all liver metastasis samples, all patient-matched primary-metastasis samples; Supplementary Fig. 1) according to the robust multi-array average approach[46] using the function justRMA in the R package affy

(v1.64.0)[47] and custom CDF files from Brainarray (hta20hsgenco-degcdf_23.0.0 and huex10sthsgencodegcdf_23.0.0). A batch effect from different lot numbers of the GeneChip™ WT Plus Reagent Kit was corrected among multiregional primary tumor samples with ComBat in the R package sva (v.3.36.0)[48] using default parameters. Gene annotations were retrieved from GENCODE using the gencode.v29.annotation.gtf file. Only protein-coding genes were retained and genes on the Y chromosome were excluded. Entrez IDs were obtained using the R package org.Hs.eg.db (v.3.10.0) and gene symbols were updated with checkGeneSymbols in HGNChelper (v.0.8.1).

Raw RNA sequencing reads were processed in a bioinformatics pipeline implemented with Snakemake (v.6.6.1) and using Python (v.3.9.5), Java (v.11.0.2) and PyYAML (v.5.4.1). The pipeline has previously been described and included adapter trimming with Trimmomatic (v.0.38), read alignment to the human reference genome GRCh38.p13 (v.41) using STAR (v.2.7.6a) with 2-pass mapping and the feature annotation file gencode.v41.annotation.gtf, quantification of reads mapping to protein-coding genes using the HTseq-count tool (v.2.0.2), and normalization of gene expression levels by estimation of transcripts per million (TPM) for non-overlapping exonic gene lengths[49]. The TPM values were log2-transformed after adding a pseudocount of 1.

## Gene expression classification and enrichment analyses

Tumor samples were classified according to CMS with the R package CMSclassifier (v.1.0.0)[8] and using the function classifyCMS.RF with a custom posterior probability threshold of 0.4. The threshold was adjusted in the multiregional primary tumor set to lower the number of unclassified samples while retaining proportionality in the number of tumors with homogeneous and heterogeneous CMS classifications (Supplementary Fig. 31). CRIS classifications were assigned with the function cris_classifier in the R package CRISclassifier (v.1.0.0)[19], using the inverse of log2-transformed gene expression data and default parameters (false discovery rate [FDR] <0.2). iCMS classification was performed using the approach and gene template described in the original publication[20]. Gene expression matrices on log2-scale were normalized to z-scores using ematAdjust and classified with the nearest template prediction approach using the function ntp and an FDR-threshold <0.05 in the R package CMScaller (v.2.0.1)[34].

Differential gene expression analyses were performed with limma as implemented in the function subDEG in CMScaller and with $p$ value adjustment by the Benjamini–Hochberg procedure. Tumor-infiltrating cancer-associated fibroblasts and cytotoxic lymphocytes were estimated using the R package MCPcounter (v.1.2.0)[50] on a combined and batch corrected gene expression dataset of all primary tumor samples ($n = 704$). Gene set enrichment analyses were performed with the R package topGO (v.2.38.1) using fisher statistics and the weight01 algorithm, as well as with the WEB-based Gene SeT AnaLysis Toolkit (WebGestalt)[51] using default settings, over-representation analysis in the Wikipathway cancer database, FDR < 0.05 and the complete list of protein-coding genes as reference. Sample group comparisons were performed with the subCamera function in CMScaller on a custom gene set collection relevant for CRC ($n = 54$; Supplementary Data 2) and with FDR adjustment of $p$ values according to the Benjamini–Hochberg procedure. One random sample from each tumor in the multiregional sample set was selected for comparisons according to CMS heterogeneity (the analysis was repeated across all multiregional samples and showed highly similar results; Supplementary Data 2). Single-sample enrichment scores were estimated with gene set variation analysis using the gsva function in R package GSVA (v.1.34.0)[52].

## Intra-tumor transcriptomic heterogeneity

For tumors with multiregional samples, intra-tumor heterogeneity was evaluated as discordant sample classifications within subtyping

frameworks (the subtype representing ≥50% of samples per tumor was considered the major component) and by general transcriptomic heterogeneity. The latter was estimated as the maximum Euclidean distance of PC1–PC3 for any pair of samples from each tumor.

For primary tumors with single samples, intra-tumor CMS heterogeneity was estimated based on enrichment scores for each CMS class (the approach is illustrated in Supplementary Fig. 6). The single-sample HTA and HuEx datasets were analyzed separately. First, template gene sets for each of the four CMS classes were identified by differential gene expression analyses comparing tumors in each class with the rest using limma (Benjamini–Hochberg adjusted $p$ value < 0.001 and log2 fold change > |1.0|; Supplementary Data 13 and 14). Second, enrichment scores for each CMS-specific template gene set in each sample were obtained using the gsva function in the R package GSVA for up-regulated genes only, and with the functions simpleScore and rankGenes in the R package singscore (v.1.6.0)[25] for up- and down-regulated genes combined. The CMS enrichment scores were evaluated in a similar analysis of the multiregional sample set, and the strongest correlations to the posterior probabilities from the original random forest CMSclassifier were found for the singscore enrichments (Spearman's $\rho$ > 0.8; Supplementary Fig. 32). Singscore also provided functions to evaluate statistical significance (generateNull and getP-vals) and was selected for further analyses. Single-sample tumors were considered unclassified if none of the four CMS enrichment scores were significant, and classified with intra-tumor CMS heterogeneity if more than one was significant ($p$ < 0.05). The approach was further evaluated in the multiregional sample set using the CMS template gene sets derived from single-sample tumors analyzed on the same platform (HTA). The major subtype of each multiregional sample was largely concordant with assignments from the original random forest CMSclassifier with an overall accuracy of 85% (Cohen's $\kappa$ = 0.77), and the majority (84%) of misclassified samples were from heterogeneous tumors (Supplementary Fig. 8a). The accuracy of computational intra-tumor CMS heterogeneity classifications (at least one sample classified as heterogeneous per tumor) was 72% relative to the spatially resolved analysis of multiregional samples (sensitivity of 73% and specificity of 68%).

## Gene-wise intra-tumor heterogeneity

Intra-tumor heterogeneity of the expression level of each protein-coding gene ($n$ = 18,823) was estimated in the multiregional sample set using a previously published method[53]. In brief, a linear mixed effects model was fitted for each gene across all samples from all tumors using the function lmer in the R package lme4 (v.1.1-29)[54] and with "tumor" as the random effect. Intra-class correlation coefficients (ICCs) were calculated for each model (gene) using the function icc in the R package performance (v.0.10.4)[55]:

$$\text{ICC} = \frac{\sigma_i^2}{\sigma_i^2 + \sigma_\epsilon^2} \tag{1}$$

Here, $\sigma_i^2$ is the random effects variance, that is, the variance explained by the grouping structure (tumor) and $\sigma_\epsilon^2$ is the residual variance. An ITH-score for each gene was calculated as:

$$\text{ITH}_{gene} = 1 - \text{ICC}_{gene} \tag{2}$$

Genes with low expression variation across the dataset (10–90th percentile range <1; $n$ = 15,585 genes) were considered non-informative and filtered out, retaining 3238 genes (17.2%) for analyses (Supplementary Fig. 11 and Supplementary Data 8). Genes were categorized according to the ITH-score using the previously published thresholds in four categories[53], or custom thresholds in the three categories ITH-low, ITH-intermediate and ITH-high (Supplementary Fig. 12 and Supplementary Data 9). The two different thresholds to score ITH-low

genes were compared in gene set enrichment analyses and showed largely concordant results (Supplementary Fig. 33). The custom threshold retained the largest number of ITH-low genes and was used for further analyses.

## Tumor classification based on ITH-low genes

Subtype discovery based on ITH-low genes was performed by the NMF approach implemented in the R package NMF (v.0.23.0)[56] using the function nmf with the brunet method[57], predefined ranks 2–10 and nrun = 100 on the inverse of log2-transformed gene expression data. The cluster number ($k$) preceding the first, large drop in the silhouette width and cophenetic score was selected as the optimal number of clusters. To evaluate a potential impact of the use of different gene expression platforms and the inclusion of multiregional samples for a subset of tumors, NMF was run both for the complete primary tumor sample set ($n$ = 704 samples from 516 tumors) and for single, randomly selected samples from each of the primary tumors analyzed on HTA ($n$ = 315). This resulted in $k$ = 5 and $k$ = 4 optimal sample clusters, respectively (Supplementary Fig. 18). There was a near perfect concordance in sample clustering between the two runs at $k$ = 4 (considering overlapping samples between the two sets only; accuracy 97%, Cohen's $\kappa$ = 0.96; Supplementary Fig. 19a). In the full sample set, the largest sample cluster from $k$ = 4 was subdivided into two clusters at $k$ = 5 (Supplementary Fig. 19b, c), but gene set enrichment analyses showed little discrimination between the two clusters (Supplementary Fig. 20). The full tumor series and NMF at $k$ = 4 was therefore used for further analyses, to strengthen the biological and statistical rigor. NMF classification was also tested using ITH-low genes defined by the previously published scoring threshold as a template (ITH-score 0–0.2; $n$ = 396 genes)[53]. This resulted in only two sample clusters differentiated mainly based on MSI/MSS-like gene expression characteristics (Supplementary Fig. 21). Classification of liver metastases by NMF was also based on genes identified as ITH-low in primary tumors. Alluvial diagrams were plotted using the R package ggalluvial (v.0.12.4).

## Classification of external tumor series based on ITH-low genes

The ITH-low gene set was filtered prior to analyses of two external tumor series, due to variation in gene expression platforms. Tumors in GSE39582 ($n$ = 566) were analyzed on Affymetrix Human Genome U133 Plus 2.0 Arrays, and probe sets were mapped to unique gene symbols using the function collapseRows in the R package WGCNA (v.1.72-5) with default settings[58]. Genes with low median expression (<4 on log2-scale) or variance (<0.1) across the tumors were filtered out, retaining 1217 (79%) of the ITH-low genes. Filtering with the same thresholds in TCGA ($n$ = 573 tumors analyzed by RNA sequencing) retained 1387 (90%) of the ITH-low genes. The two tumor series were classified separately according to the same approach as in the in-house series, using NMF with predefined ranks 2–6 on the filtered set of ITH-low genes. Pathway enrichment analysis of differentially expressed genes between NMF subtypes (limma: Benjamini–Hochberg adjusted $p$ value < 0.001 and log2 fold change > |1.0|) was performed with the R package pathfindR (v.2.3.0) using default settings, including testing of the KEGG pathway database and Benjamini–Hochberg adjustment of $p$ values[59].

## Statistical analyses

All statistical analyses were performed in R v.4.2.2[60]. Two-sided $p$ values < 0.05, or adjusted $p$ values as specified, were considered significant. Principal components analysis was performed with the prcomp function in the package stats (v.4.2.2) based on the genes with highest cross-sample variance ($n$ = 1000). Pearson's and Spearman's correlations were calculated and visualized using the functions cor, cor.mtest and corrplot in the R package corrplot (v.0.92), and with conf.level = 0.95. Odds ratios and 95% CIs were estimated with Fishers' exact test (fisher.test), and were together with Pearson's chi-squared

test (chisq.test) and Welch's two sample *t*-test (t.test) used to evaluate associations between clinicopathological parameters and sample groups according to molecular characteristics. Classification accuracy and Cohen's κ were estimated with the function confusionMatrix in the package caret (v.6.0-93). The center line of box plots represents the median, boxes represent the interquartile range, and whiskers represent 1.5× the interquartile range above the 75th percentile (maxima) or below the 25th percentile (minima).

Survival analyses were performed for patients with stage I–III CRC (unless otherwise stated) treated by complete tumor resection (residual tumor status R0) and with no pre-surgical chemoradiation or synchronous tumors. Five-year RFS was the primary endpoint and estimated as time from surgery to relapse or death from any cause. Patients with no events were censored after 5 years or at last follow-up. Overall survival was evaluated as the time from surgery to death from any cause. Multivariable and univariable Cox proportional hazards models were estimated using the coxph function in the survival package (v.3.4-0) with *p* values from Wald tests. The proportional hazards assumption was assessed for all models using the cox.zph function, and all variables met the assumption, except for patient age or *KRAS* mutation status in multivariable models including gene expression subtypes. Stratification of models according to these variables did not have a strong impact on the results (Supplementary Data 6 and 11). Kaplan–Meier plots were generated with the ggsurvplot function in the survminer package (v.0.4.9), with *p* values from Wald test. The proportion of explained variation in 5-year RFS by each variable in multivariable models was calculated using rsq in the survMisc package (v.0.5.6)[61], and bootstrapped with 5000 iterations and sampling with replacement. Survival analyses of the GSE39582 series were also performed for patients with stage I–III cancers and with 5-year RFS as the endpoint (*n* = 493 patients with follow-up data). Survival analysis of the TCGA series was not performed due to short follow-up time of the majority of patients (70% were lost to follow-up during the first 12 months).

### Reporting summary

Further information on research design is available in the Nature Portfolio Reporting Summary linked to this article.

## Data availability

The microarray gene expression data are publicly available. Multi-regional primary CRC samples generated in this study (*n* = 286) have been deposited to the NCBI's Gene Expression Omnibus under accession code GSE241101. The extended single-sample primary tumor set has previously been deposited under accession codes GSE24550, GSE29638, GSE69182, GSE79959, GSE139170, and GSE96528. The liver metastases samples have previously been deposited under accession code GSE159216. The raw RNA sequencing data are considered patient identifiable and subject to secure storage regulations in accordance with Norwegian legislation and the ethical approval of the study by the Regional Committee for Medical and Health Research Ethics, South Eastern Norway (data will be made available upon request to the corresponding author at email address anita.sveen@rr-research.no, and this will require formalization of a data transfer agreement). Public gene expression data from the GSE39582 and TCGA series were downloaded from the SAGE Bionetworks Synapse platform [https://www.synapse.org/#!Synapse:syn2634724], and single-cell RNA sequencing data were downloaded from GEO under accession code GSE144735. Source data are provided with this paper.

## Code availability

All data processing and analyses were performed with published software packages and computer code, and have been described and cited in the "Results" and/or "Methods". No custom code was developed in the study.

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

## Acknowledgements

The study was financially supported by the South-Eastern Norway Regional Health Authority (project numbers 2019042 and 2023101 to A.S. and 2017102 and 2021058 to R.A.L.), the Research Council of Norway (project number 287899 to A.S. and 250993 [FRIPRO Toppforsk] to R.A.L.), and the Norwegian Cancer Society (project number 208336-2019 to A.S. and 223319-2021 to R.A.L.). RNA sequencing and library generation was performed at the Oslo University Hospital Genomics Core Facility.

## Author contributions

Study conception and design: J.L., A.N., R.A.L. and A.S. Data acquisition: J.L., I.A.E., S.H.M., I.F.B., M.H., O.H.S., A.N., R.A.L. and A.S. Data analysis and interpretation: J.L., S.M.K.K., H.M.R., M.J., S.T., R.A.L. and A.S. First manuscript draft: J.L. and A.S. Study supervision: A.S. All authors were involved in revision of the manuscript and have approved the final version.

## Competing interests

A.N., R.A.L. and A.S. are co-inventors of a patent application regarding the use of HSP90 inhibitors in relation to the consensus molecular subtypes of colorectal cancer (PCT/IB2018/000042). S.H.M., R.A.L. and A.S. are co-inventors of a patent application describing transcriptomic liver metastasis subtypes (LMS) of colorectal cancers (Attorney Docket No. INVEN-39613.101). The authors declare that they have no other competing interests.
