## [Peer Review File · Nature Communications]

Multiregional transcriptomics identifies congruent consensus subtypes with prognostic value beyond tumor heterogeneity of colorectal cancerREVIEWER COMMENTS

Reviewer #1 (Remarks to the Author): expertise in colorectal cancer transcriptomics

In the study from Langerud et al, the authors present a new classification approach for colorectal cancer that looks to circumvent the issues of intra-tumoural heterogeneity, with particular emphasis on in this study on CMS and iCMS.

Utilising multiregional primary tumour transcriptomics from CRC as well as primary-liver met matches samples, the authors explored CMS classification and highlighted that the heterogeneity in tumours were in large part contributed from stromal infiltration and these heterogeneous tumours were inclined to relate with poor clinical outcome. Using the ITHlow genes, authors went on to develop a new/alternative classification called congruent CMS where they show that iCMS are largely concordant with their subtyping. The authors highlight the importance of the congruent CMS in terms of prognosis where it is less undermined by intra-tumour heterogeneity than that of CMS, despite portraying similar biological characteristics.

Overall, this is an excellent study, detailed in a well-written manuscript with a clarity of figures that is commendable. There is substantial supplementary information provided, all of which supports the conclusions here, and the authors have done a really good job on ensuring that the narrative is clearly articulated and represented in the main figures. This timely study will certainly advance the field, and provides a clear example of the value that can still be derived from using bulk tumour samples for discovery, in a field that has become somewhat obsessed with the use of single cell data (and its highly manipulated extractions processed) as the sole route for biological discovery. Indeed, this is demonstrated by how well aligned this bulk approach identifies the iCMS approach.

The generation of a "stable" classifier has long been needed to robustly classify tumour samples, however the stability can also come at a cost, as it can remove precisely the biological dynamics and variability that may drive the aggressive outcomes that are demarked underneath the stable biomarker. As such, while iCMS and K2 are definitively showing a stable set of 2 epithelial subtypes here, the overlaying of biological cascades (that we know vary) will add more to this going forward. I look forward to seeing the authors progress on this eloquent work in future studies.

My comments and suggestions below should be seen as points that can hopefully add to this already well structure work, none of which should preclude eventual acceptance of this study.

Minor points:

Point 1 - Lines 84 – 87 The most common CMS combinations were CMS2/4 (n = 19, 49% of heterogeneous tumors) and CMS1/3 (n = 8, 21%), with CMS4 and CMS3 as the minor components, respectively. Combinations of CMS3/4 (n = 3, 8%) and CMS1/2 (n = 2, 5%) were rare.

Would be really useful to actually see examples of some of these lesions, as a WSI of H&E cryosections with annotated regions from where the samples were taken from (if possible) or else individual images of these regions. The detail in M&M describes them as "spatially distinct" however a bit more detail on what this involved would be good – for the multiregional ones were these tumour samples that had regional biopsies taken after gross resection - or other? However I do appreciate that access to sample images may not be possible.

Point 2 – Survival analyses. Some of these include stage I-III, it might be useful to have an additional analysis where stage I is excluded, given the relatively few events that this group will add. Similarly, an accompanying supplementary figure could assess the 0.4 v 0.5 threshold used here for CMS.

Point 3 - Figure 3B, have you looked into intestinal stems in more granularity (for example, with CSC/RSC from <https://doi.org/10.1016/j.stem.2022.07.008>) as in ITH-high genes are highly correlated with "cancer stem cells (LGR5)" which mostly should align with epithelial CMS2 tumour so thought it would correlate with ITH-low genes rather than ITH-high? Would be useful to check it assessment of the opposing stem class reveals an opposing result here.

Point 4 - Figure S3: Interesting observation! While CMS4 should be associated with mesenchymal-like traits, the figure shows that in homogeneous vs heterogeneous in CMS4, both cancer mesenchymal transition and epithelial mesenchymal transition does not appear solely in CMS4. Moreover, MYC targets (and similar genesets) seems to be enriched in heterogeneous within CMS4. It would be interesting to know what the authors feel underpins this result (no need for new analyses, but a suggestions as to what mechanism and/or genes in these geneset could be contributing to the heterogeneity of the tumour.)

Point 5 - Figure S25: When comparing the features between cCMS, we know that CMS2 are strongly associated with MYC targets, G2M, proliferative features, but the new classification show cCMS1 are much more enriched for these features than cCMS2. A comment from the authors as to why this may be would be useful, and also to highlight that "hallmarks" of a large subtype like CMS2 may be misleading when you look at the inevitable sub-subgroups that lie within

Again, I commend the authors on the clarity of their work and the approaches they have taken here.

Reviewer #2 (Remarks to the Author): expertise in bioinformatics clustering and evolution analysis

The authors performed multiregion RNA-seq to demonstrate the presence of heterogeneous subtypes, which have different CMS subtypes in different regions. The heterogeneous subtypes are associated with worse prognosis. They identified genes that show variations across different regions and the ones with less variation. Finally, they proposed a new framework for subtyping, based on the gene with less with less variation and present relationships with previously proposed subtyping frameworks. The story of this work is interesting, but I have an impression that more data is necessary for publication. At least following points should be addressed.

1. Do you find Any genomic correlate associated with subtype heterogeneity? In Figure 1 A MSI and RAS/BRAF status are indicated for each case. Could you present the data for multiple samples for each case? Can you infer chromosomal copy number alteration from expression data and check heterogeneity of it?
2. The proposed new framework is evaluated only on the authors' own dataset. It should be evaluated extensively using publicly available independent datasets.

Reviewer #3 (Remarks to the Author): expertise in bioinformatics analysis of cancer evolution

This paper studies whether multiregional bulk RNA sequencing can lead to more accurate

transcriptomic profiling compared to the use of a single bulk sample per tumor. The paper finds that current CMS classifications may vary among spatial samples from the same tumor. The paper concludes by deriving new CMS signatures using NMF restricted to ITH-low genes (genes that have similar variability among all multiregional samples of the same tumor). I have several comments.

- Do the new CMS signatures perform better in survival analysis compared to previous CMS signatures?

- It is not clear why there is a distinction between the $k=2$ and $k=4$ signatures, especially when the $k=2$ signatures are subsumed by the $k=4$ signatures. It would be good to stick with only one set of signatures.

- This was unclear to me:

"To extend the analyses to a larger patient series, we used CMS as a framework for computational modeling of heterogeneity in single, bulk tissue samples from another 418 primary CRCs (Supplementary Table S1)"

It seems to correspond to the approach described in lines 465-479. If so, please include more details in the main text. Also, discuss validation on multiregional samples. A proper validation on the multiregional samples would entail a training/test split to determine accuracy of classification.

- It would be helpful to discuss how the original CMS types themselves were inferred. What if they were confounded by heterogeneity, or by tumor purity?

RESPONSE TO REVIEWERS' COMMENTS

Authors' responses are highlighted in blue below each reviewer comment. Please note that page numbers refer to the revised manuscript with tracked changes.

Reviewer #1 (Remarks to the Author): expertise in colorectal cancer transcriptomics

In the study from Langerud et al, the authors present a new classification approach for colorectal cancer that looks to circumvent the issues of intra-tumoural heterogeneity, with particular emphasis on in this study on CMS and iCMS.

Utilising multiregional primary tumour transcriptomics from CRC as well as primary-liver met matches samples, the authors explored CMS classification and highlighted that the heterogeneity in tumours were in large part contributed from stromal infiltration and these heterogeneous tumours were inclined to relate with poor clinical outcome. Using the ITHlow genes, authors went on to develop a new/alternative classification called congruent CMS where they show that iCMS are largely concordant with their subtyping. The authors highlight the importance of the congruent CMS in terms of prognosis where it is less undermined by intra-tumour heterogeneity than that of CMS, despite portraying similar biological characteristics.

Overall, this is an excellent study, detailed in a well-written manuscript with a clarity of figures that is commendable. There is substantial supplementary information provided, all of which supports the conclusions here, and the authors have done a really good job on ensuring that the narrative is clearly articulated and represented in the main figures. This timely study will certainly advance the field, and provides a clear example of the value that can still be derived from using bulk tumour samples for discovery, in a field that has become somewhat obsessed with the use of single cell data (and its highly manipulated extractions processed) as the sole route for biological discovery. Indeed, this is demonstrated by how well aligned this bulk approach identifies the iCMS approach.

The generation of a "stable" classifier has long been needed to robustly classify tumour samples, however the stability can also come at a cost, as it can remove precisely the biological dynamics and variability that may drive the aggressive outcomes that are demarked underneath the stable biomarker. As such, while iCMS and K2 are definitively showing a stable set of 2 epithelial subtypes here, the overlaying of biological cascades (that we know vary) will add more to this going forward. I look forward to seeing the authors progress on this eloquent work in future studies.

My comments and suggestions below should be seen as points that can hopefully add to this already well structure work, none of which should preclude eventual acceptance of this study.

Minor points:

Point 1 - Lines 84 – 87 The most common CMS combinations were CMS2/4 (n = 19, 49% of heterogeneous tumors) and CMS1/3 (n = 8, 21%), with CMS4 and CMS3 as the minor components, respectively. Combinations of CMS3/4 (n = 3, 8%) and CMS1/2 (n = 2, 5%) were rare.

Would be really useful to actually see examples of some of these lesions, as a WSI of H&E cryosections with annotated regions from where the samples were taken from (if possible) or else

individual images of these regions. The detail in M&M describes them as "spatially distinct" however a bit more detail on what this involved would be good – for the multiregional ones were these tumour samples that had regional biopsies taken after gross resection - or other? However I do appreciate that access to sample images may not be possible.

Authors' response: Whole slide images of the tumors were unfortunately not available; however, we have performed H&E staining of cryosections of the individual regions of three selected example tumors with CMS heterogeneity. The stains were performed on neighboring sections of the samples used for RNA extractions. The images are included in new Supplementary Figures S2C-E. These illustrate morphological differences according to CMS classification (as previously described by Sirinukunwattana et al. Gut 2021;70:544-54).

The multiregional samples were taken from specimens after major surgical resection. Following our routine sampling protocol, tumors were sampled at the operating theatre and prior to pathological examination. This has been specified in the Patient material description of the revised manuscript (page 20). We have to make sure not to compromise the surgical specimens for diagnostics, and can unfortunately not provide a detailed spatial map of the collected regions. However, multiple samples are not taken unless we are certain that the sampled regions are clearly spatially separated. Tumor size is therefore a critical factor. Tumors with multiregional samples had a diameter of minimum 15 mm (median diameter 40 mm, 95% CI 35-40). This has also been included in the Patient material description (page 20), and is illustrated relative to the number of sampled regions in Supplementary Figure S2A. Tumor size was not significantly associated with the number of sampled regions, indicating that the analyses are not biased by tumor size.

Point 2 – Survival analyses. Some of these include stage I-III, it might be useful to have an additional analysis where stage I is excluded, given the relatively few events that this group will add. Similarly, an accompanying supplementary figure could assess the 0.4 v 0.5 threshold used here for CMS.

Authors' response: We appreciate the suggestions and have added the following Supplementary Figures showing that results from survival analyses remained similar when excluding patients with stage I tumors, and when using the default posterior probability threshold for CMS classification (0.5):

- Supplementary Figure S9A: Kaplan-Meier curves of five-year RFS according to intra-tumor CMS heterogeneity, and CMS heterogeneity stratified by CMS4 classification among patients treated by complete resection of stages II and III CRC. These plots correspond to the main Figures 2A and 2C.
- Supplementary Figure S27B: Kaplan-Meier curves of five-year RFS according to cCMS classification among patients treated by complete resection of stages II and III CRC. The figure also includes a pie chart illustrating the proportion of explained variation in five-year RFS by each variable in a multivariable Cox proportional hazards model (of stage II+III). These plots correspond to the main Figures 5D and 5E.
- Supplementary Figure S28: Kaplan-Meier plots of five-year RFS according to the original CMS classification among patients with stage I-III CRC, using posterior probability thresholds of 0.4 and 0.5.

Point 3 - Figure 3B, have you looked into intestinal stems in more granularity (for example, with CSC/RSC from <https://doi.org/10.1016/j.stem.2022.07.008>) as in ITH-high genes are highly correlated with “cancer stem cells (LGR5)” which mostly should align with epithelial CMS2 tumour so thought it would correlate with ITH-low genes rather than ITH-high? Would be useful to check it assessment of the opposing stem class reveals an opposing result here.

Authors’ response: This is an interesting question and we have addressed this with additional analyses. Figure 1 below shows the Spearman correlations (absolute values) of all stem cell signatures with PC1 or PC2 of ITH-high and ITH-low genes, including the CBC and RSC signatures proposed by the reviewer. Both the CBC and RSC signatures are indeed most strongly correlated to PC1 of ITH-low genes (with opposite signs). However, the correlations are not sufficiently strong to be included for plotting in Figure 3B (thresholds indicated by vertical dashed lines in the figure below). As expected, the CBC signature correlated to the base of colonic crypts signature (Pearson correlation 0.75), which is predominantly ITH-low and included in Figure 3B, and the RSC signature correlated to the top of colonic crypts signature (Pearson correlation 0.67).

Figure 1. Spearman’s correlation coefficients (absolute values) between PC1 or PC2 of ITH-high (right) or ITH-low (left) genes and single-sample enrichment scores (GSVA) of stem cell signatures. Vertical dashed lines indicates thresholds to rank among the top-15 gene sets included for plotting in Figure 3B in the manuscript.

To provide a potential rationale for the correlations between the LGR5 and EPHB2 cancer stem cell signatures with PC1 of ITH-high genes, we have investigated the distribution of ITH-scores among genes of each gene set. These distributions largely supported the correlation analyses, showing low ITH-scores of MSI/MSS and cell cycle-related genes relative to epithelial-mesenchymal transition genes. Genes of WNT/β-catenin signaling and the several stem cell signatures were predominantly ITH-low. However, a small subset of genes in the LGR5 and EPHB2 cancer stem cell signatures were ITH-high (the identity of these genes is provided in the Source Data file). We reason that this likely contributed to the correlation of these signatures to PC1 of ITH-high genes. Similarly, genes involved in hedgehog signaling showed a near bimodal distribution of ITH-scores, and this likely accounted for the correlation of this gene set with both PC1 of ITH-high genes and PC2 of ITH-low genes. We consider these data to provide a more nuanced description of the associations between gene sets and the ITH-scores, and have included the analyses in a new Figure 3D in the revised manuscript. A paragraph describing the data has also been added to the Results section (pages 9-10).

The original CMS publication (Guinney et al., Nat Med 2015;21:1350-6) also showed enrichment with cancer stem cell signatures in mesenchymal-like CMS4 tumors, which is the subtype with largest intra-tumor heterogeneity in our study. This is consistent with a cellular plasticity model and with

dedifferentiation from epithelial-like and lineage committed cells to mesenchymal-like cells with stem cell properties.

Point 4 - Figure S3: Interesting observation! While CMS4 should be associated with mesenchymal-like traits, the figure shows that in homogeneous vs heterogeneous in CMS4, both cancer mesenchymal transition and epithelial mesenchymal transition does not appear solely in CMS4. Moreover, MYC targets (and similar genesets) seems to be enriched in heterogeneous within CMS4. It would be interesting to know what the authors feel underpins this result (no need for new analyses, but a suggestions as to what mechanism and/or genes in these geneset could be contributing to the heterogeneity of the tumour.)

Authors' response: We have included a new Supplementary Figure S3B with boxplots of single-sample enrichment scores (GSVA scores) of selected signatures to shed light on this question. The boxplots compare CMS4 samples from tumors with heterogeneous versus homogeneous CMS4 classifications, and show that signatures of mesenchymal-like traits and stromal infiltration were high in all CMS4 samples, regardless of the classification of other regions of the tumor. This illustrates why the signatures were not enriched in group comparisons of heterogeneous and homogeneous CMS4 tumors. In contrast, the signatures of MYC targets and the proliferative colonic crypt had high scores in CMS4 samples from a subset of tumors with a major CMS2 component (CMS2:CMS4 tumors; both signatures were high in the same subset of samples). This indicates that there was an admixture of CMS2 also in the samples classified as CMS4, and the intra-tumor heterogeneity was reflected in intra-sample heterogeneity. These data are described also in the Results section of the revised manuscript (page 6).

Point 5 - Figure S25: When comparing the features between cCMS, we know that CMS2 are strongly associated with MYC targets, G2M, proliferative features, but the new classification show cCMS1 are much more enriched for these features than cCMS2. A comment from the authors as to why this may be would be useful, and also to highlight that "hallmarks" of a large subtype like CMS2 may be misleading when you look at the inevitable sub-subgroups that lie within

Authors' response: We have included new Supplementary Figures S25B-C to illustrate more details of the data underlying this apparent ambiguity. Samples that were cCMS2 but non-CMS2 had low signature scores for MYC targets (and the cell cycle signatures). In contrast, a subset of the samples that were cCMS1 but non-CMS1 had high MYC target scores. Notably, cCMS1-nonCMS1 samples and cCMS2-nonCMS2 samples were more frequently from tumors with CMS heterogeneity than samples that showed corresponding cCMS1-CMS1 and cCMS2-CMS2 classifications. This highlights that heterogeneity of the subtypes, tumors and/or samples indeed likely contributed to discordant enrichments for the MYC target signature in the cCMS versus original CMS framework. No enrichment with heterogeneous tumors was seen according to cCMS3/CMS3 or cCMS4/CMS4 classifications, likely contributing to the highly correspondent enrichments for these classes. We have included additional text in the Results to highlight these data (page 13).

In response to comment 2 from Reviewer 2, we have also performed new validation analyses by subtype discovery of two external primary tumor series based on the ITH-low genes (the GSE39582 and TCGA series). High scores for MYC targets and cell cycle signatures in the cCMS1-corresponding class were supported in both the external series. Furthermore, potential for subclassification of the large and heterogeneous CMS2-corresponding class was consistently indicated in all three tumor

series (in-house and two external). The two subtypes were distinguished by characteristics such as bacterial and viral infection, as well as several oncogenic or tumor suppressor signaling pathways. These analyses have been described in a new paragraph of the Results section (pages 14-16), a new Figure 6 and Supplementary Figures S29 and S30, as well as in the Discussion (page 18).

Again, I commend the authors on the clarity of their work and the approaches they have taken here.

We thank the reviewer for the positive remarks and for the clever suggestions that we have addressed above.

Reviewer #2 (Remarks to the Author): expertise in bioinformatics clustering and evolution analysis

The authors performed multiregion RNA-seq to demonstrate the presence of heterogeneous subtypes, which have different CMS subtypes in different regions. The heterogeneous subtypes are associated with worse prognosis. They identified genes that show variations across different regions and the ones with less variation. Finally, they proposed a new framework for subtyping, based on the gene with less with less variation and present relationships with previously proposed subtyping frameworks. The story of this work is interesting, but I have an impression that more data is necessary for publication. At least following points should be addressed.

1. Do you find Any genomic correlate associated with subtype heterogeneity? In Figure 1 A MSI and RAS/BRAF status are indicated for each case. Could you present the data for multiple samples for each case? Can you infer chromosomal copy number alteration from expression data and check heterogeneity of it?

Authors' response: We would like to point out that the manuscript does describe results from association analyses of CMS heterogeneity with MSI status, *KRAS/NRAS* mutations and *BRAF*^{V600E} mutations:

- All the genomic markers were indeed scored in all samples from tumors with CMS heterogeneity (this is stated in the “MSI and mutation analyses” section of the Methods, page 22), and all the markers were concordant among the multiregional samples (stated in the first section of the Results, on page 5).
- There was no enrichment with any of the genomic markers in tumors with heterogeneous versus homogeneous CMS classifications. This is shown in Supplemental Table S5.
- Associations between the genomic markers and specific CMS classes were also analyzed in the context of intra-tumor heterogeneity, and this revealed new details of previously known associations (the data and corresponding statistics are described in the first section of the Results, on page 5):
 - MSI and *BRAF*^{V600E} mutations were strongly enriched among tumors with a major CMS1 component (tumors with > 50% CMS1 samples), and were not similarly frequent in CMS1-minor tumors (tumors with < 50% CMS1 samples).
 - *KRAS/NRAS* mutations were most frequent in CMS3 tumors without a CMS1 component.

We appreciate that bioinformatics tools to infer copy number variants from RNA sequencing data have been developed (for example, based on features such as the minor allele frequency). However, the multiregional samples were analyzed using high-resolution microarrays, not by RNA sequencing, and corresponding bioinformatics approaches are not readily available for microarray data. We appreciate that comparisons of transcriptomic heterogeneity with large-scale genomics data could provide additional insights, but such data are unfortunately not available.

2. The proposed new framework is evaluated only on the authors' own dataset. It should be evaluated extensively using publicly available independent datasets.

Authors' response: We appreciate this request and have included validation analyses of the two largest publicly available datasets of primary colorectal tumors, including GEO accession number GSE39582 (Marisa et al., PLoS Med 2013;10:e1001453) and TCGA (The Cancer Genome Atlas Network, Nature 2012;487:330-337). Processed gene expression data and metadata were downloaded from the SAGE Bionetworks Synapse platform, including 566 tumors from GSE39582 and 573 tumors from TCGA (<https://www.synapse.org/#!/Synapse:syn2634724>). An overview of clinicopathological parameters and molecular markers of the external datasets has been included in a new Supplementary Table S12.

We performed validation analyses by tumor classifications according to the same approach as in the in-house series. That is, subtype discovery was performed by unsupervised NMF based on ITH-low genes, rather than by development of a cCMS prediction model. The approach is described in a new paragraph of the Methods section, including filtering of the ITH-low genes due to use of different gene expression platforms (page 28). Results from the validation analyses are described in a new paragraph of the main text on pages 14 to 16 ("ITH-low classifications of external primary tumor series"), as well as in a new main Figure 6 and new Supplementary Figures S29 and S30.

The validation analyses supported that the ITH-low genes distinguished tumors according to the same biological and clinicopathological characteristics in independent tumor series. However, different factorization ranks (number of sample clusters/ k in NMF) were needed to identify corresponding subtypes in the different series. We therefore acknowledge in the Discussion that the optimal number of ITH-low subtypes remains inconclusive (page 18). In particular, a potential for subclassification of the large and heterogeneous group of CMS2-corresponding tumors based on characteristics such as bacterial or viral infections was supported in all three tumor series (in-house and two external validation sets). This subtype was consistent with a microbiome-dependent subtype proposed in a previous study (Bramsen J et al., Cell Rep 2017;19:1268-1280), and provides added value from the validation analyses to our study. Survival analyses among stage I-III cancers in GSE39582 also supported an added prognostic value of the ITH-low subtypes relative to the original CMS. Survival analyses of TCGA were not performed due to short follow-up time of the majority of patients (70% were lost to follow-up during the first 12 months), and this has been described in the Methods section (page 29).

Reviewer #3 (Remarks to the Author): expertise in bioinformatics analysis of cancer evolution

This paper studies whether multiregional bulk RNA sequencing can lead to more accurate transcriptomic profiling compared to the use of a single bulk sample per tumor. The paper finds that current CMS classifications may vary among spatial samples from the same tumor. The paper concludes by deriving new CMS signatures using NMF restricted to ITH-low genes (genes that have similar variability among all multiregional samples of the same tumor). I have several comments.

- Do the new CMS signatures perform better in survival analysis compared to previous CMS signatures?

Authors' response: The prognostic value of the original CMS classification is largely explained by tumor microenvironment components, and CMS has no/little prognostic effect in multivariable models incorporating tumor-infiltrating cytotoxic lymphocyte scores and cancer-associated fibroblasts (in addition to clinicopathological factors, MSI status and *KRAS* and *BRAF*^{V600E} mutations). This has been shown in a multicenter study of more than 2,600 stage II/III CRCs (Dienstmann R, Villacampa G, Sveen A et al., *Ann Oncol* 2019;30:1622-1629), and forms part of the rationale for the current study. Furthermore, the original CMS classification is vulnerable to intra-tumor heterogeneity and sampling bias, also this due in particular to variation in the tumor microenvironment. This rationale is described and documented in the Introduction (page 3) with reference to relevant studies.

Figure 2 and Supplemental Table S6 of this manuscript show that intra-tumor heterogeneity of CMS has independent prognostic value in a multivariable model with tumor microenvironment components (similar to the multivariable model referred to above). Furthermore, the new and congruent CMS classification based on ITH-low genes explained a relatively large proportion of variation in five-year RFS in the same multivariable model, incorporating both intra-tumor CMS heterogeneity and the tumor microenvironment components. This is shown in Figures 5D-E, Table 1 and Supplemental Table S11. The original CMS classification was not prognostic in these patients (or in the multivariable model incorporating intra-tumor heterogeneity). This is shown in Supplementary Figure S28. We have also performed new validation analyses of an external primary tumor series (GSE39582), which supported that classification based on ITH-low genes can improve the prognostic stratification of tumors relative to the original CMS classification. These new data are shown in Supplementary Figure S29D. We therefore think that a superior prognostic value of the new classification is well documented relative to the original CMS classification in the manuscript.

- It is not clear why there is a distinction between the k=2 and k=4 signatures, especially when the k=2 signatures are subsumed by the k=4 signatures. It would be good to stick with only one set of signatures.

Authors' response: Both the k2 and k4 classifications serve an important purpose in the study.

The k2 classification was a predefined rank, with the specific purpose to compare with the two intrinsic CMS classes (iCMS) recently identified from single-cell RNA sequencing of the malignant epithelial compartment of CRCs (Joanito et al., *Nat Genet* 2022;54:963-75). The strong correspondence of the k2 classification with iCMS demonstrated that the approach taken in this

study to identify genes with low intra-tumor heterogeneity among multiregional samples could recapitulate results from the single-cell sequencing. This both supported robustness of the approach and indicated that the ITH-low genes represented cancer cell-intrinsic features. In addition, the k2/iCMS classification was used to show subtype-switching and pronounced cancer cell plasticity between patient-matched primary tumors and liver metastases (even with a two-state classification; Figure 4C).

The best model fit for classification based on ITH-low genes was k4 or k5 (depending on the input set of samples, as illustrated in Supplementary Figure S18). k5 included two clusters with highly similar characteristics (Supplementary Figures S19-S20), and k4 was therefore used for further analyses. This showed potential for further sub-classification of tumors based on cancer cell-intrinsic traits, beyond the two-state k2/iCMS classification. This represents an important additional value of our study, and of the approach of using multiregional samples (relative to single-cell sequencing). Indeed, the k4 classification allowed for improved prognostic stratification of patients in the context of intra-tumor heterogeneity. We firmly believe that both the k2 and k4 classifications are pertinent to the study. Notably, there is some ambiguity in what represents the optimal number of ITH-low subtypes, based on additional validation analyses of two external tumor series performed in this revision (new main Figure 6 and Supplementary Figures S29 and S30). We have acknowledged this uncertainty in the Discussion (on page 18).

- This was unclear to me:

"To extend the analyses to a larger patient series, we used CMS as a framework for computational modeling of heterogeneity in single, bulk tissue samples from another 418 primary CRCs (Supplementary Table S1)"

It seems to correspond to the approach described in lines 465-479. If so, please include more details in the main text. Also, discuss validation on multiregional samples. A proper validation on the multiregional samples would entail a training/test split to determine accuracy of classification.

Authors' response: The reviewer is correct that this paragraph of the Methods section describes the approach. We appreciate that inclusion of some of these details will improve the understanding of the Results description, and have included a summary on pages 6-7. We have also made small changes to the illustration in Supplementary Figure S6 to clarify the approach.

It is unfortunately not quite clear to us which validation analyses in multiregional samples the reviewer refers to. The computational intra-tumor CMS heterogeneity scores of tumors with single samples were evaluated by comparisons with the observed heterogeneity among multiregional samples. The multiregional sample set and single-sample series are from independent sets of tumors, and additional splitting in training/test sets is therefore not needed to obtain an independent validation. Further validations were performed by estimation of computational CMS heterogeneity scores also in each sample of the multiregional sample set. Please note that these heterogeneity scores were based on template gene sets obtained from the single-sample series (the gene sets listed in Supplementary Table S13), and therefore also represent independent validation analyses. In the revised manuscript we have included a new Supplementary Figure S8A illustrating the direct comparison of the computational approach to estimate intra-sample heterogeneity in the multiregional samples with the spatially resolved analysis of the same tumors. The tumors were

considered heterogeneous according to the computational approach if at least one sample was heterogeneous (that is, statistically enriched with more than one CMS template gene set). Notably, tumors with different (homogeneous) CMS classes in different spatially distinct samples were not considered heterogeneous in this validation analysis. This was to disregard the spatial resolution and to specifically evaluate the power of the computational intra-sample analysis to detect CMS heterogeneity (to mimic the analysis of tumors with availability of a single sample only). This analysis showed 72% accuracy for classification of CMS heterogeneity relative to the evaluation of multiregional samples. This limited analytical discriminatory power was a contributing factor to the lower frequency of CMS heterogeneity detected in the single-sample tumor series relative to the multiregional series (30% versus 40%, respectively). This is acknowledged in the Results section on page 7, with reference to the new Supplementary Figure S8A.

- It would be helpful to discuss how the original CMS types themselves were inferred. What if they were confounded by heterogeneity, or by tumor purity?

Authors' response: The original CMS classification was developed from bulk transcriptomics data of single samples of each tumor. This has been clarified in the Introduction on page 3 of the revised manuscript. Indeed, the classification is vulnerable to tumor heterogeneity and sampling bias, in particular due to variation in the tumor microenvironment. This is already described in the Introduction, with reference to the most relevant studies. In fact, intra-tumor heterogeneity of the original CMS was a main rationale for the current study, and the aim was to investigate the prognostic impact of heterogeneity and the potential for a transcriptomic classification less vulnerable to heterogeneity. This is also stated in the Introduction. Our results regarding the frequency and prognostic impact of intra-tumor CMS heterogeneity are put into context of the relevant literature in the Discussion on page 18. This feature of the original CMS classification is in our opinion adequately described in the manuscript.

REVIEWERS' COMMENTS

Reviewer #1 (Remarks to the Author):

The authors have presented a revised manuscript that fully addresses and in some cases goes above and beyond all my original points. This is an excellent study with substantial data included, and I again thank the authors for the clear narrative throughout.

I fully recommend publication.

Reviewer #2 (Remarks to the Author):

We appreciate that author's much efforts for revision. especially, authors added analysis using other cohorts for validate cCMS subtyping, which is a highlight of the paper. However, the added data provides me an impression that that cCMS subtyping is not robust. I am not confident that this conclusion from the paper is acceptable.

Reviewer #3 (Remarks to the Author):

My previous comments have been addressed to my satisfaction.

I reviewed the comments made by Reviewer 2. I think the reviewer raises a good point about the importance of carefully selecting the number k of clusters, which seems to be dataset specific. I think some more discussion about this point might be warranted.

RESPONSE TO REVIEWERS' COMMENTS

Reviewer #1 (Remarks to the Author):

The authors have presented a revised manuscript that fully addresses and in some cases goes above and beyond all my original points. This is an excellent study with substantial data included, and I again thank the authors for the clear narrative throughout.

I fully recommend publication.

Authors' response: We thank the reviewer for the positive remarks.

Reviewer #2 (Remarks to the Author):

We appreciate that author's much efforts for revision. especially, authors added analysis using other cohorts for validate cCMS subtyping, which is a highlight of the paper. However, the added data provides me an impression that that cCMS subtyping is not robust. I am not confident that this conclusion from the paper is acceptable.

*Authors' response: We appreciate this concern and acknowledge the need for use of different factorization ranks to identify corresponding subtypes in the different tumor series. We have already acknowledged in the Discussion that the optimal number of ITH-low subtypes remains inconclusive, and that additional studies are needed to consolidate the ITH-low classification. We have also removed the specification of *four* ITH-low classes in the Abstract and Discussion of the revised manuscript. It is not clear whether the inconsistent number of clusters is related to biological differences among the tumor series or to technical variation from use of different gene expression platforms in the three series. We have acknowledged the potential for technical bias in the Discussion of the revised manuscript.*

Nonetheless, we would like to point out that the biological attributes of the ITH-low subtypes were highly similar in all three patient series, and subtypes corresponding to the four cCMS classes were identified in both external series. Furthermore, the validation analyses support the potential to subclassify the CMS2-corresponding cluster. This was initially indicated in the in-house dataset (k=4 and k=5 had similar silhouette widths and cophenetic scores in the initial analyses), but due to subtle differences between two of the k5 clusters, k4 was chosen for presentation (as described in the manuscript). However, the k5 clusters in the in-house dataset were highly similar to the k5 clusters in GSE39582, and more detailed pathway enrichment analyses showed that the two subdivided CMS2-corresponding clusters were distinguished based on characteristics such as bacterial or viral infection and cell cycle progression in all three series analyzed. This subclassification is also consistent with a microbiome-dependent subtype proposed in a previous study (Bramsen et al., Cell Rep 2017;19:1268-80), providing further support of the ITH-low classification.

In summary, we acknowledge the need for a consensus beyond this study with respect to the number of ITH-low sample clusters, but robustness of the approach is supported by the similar molecular and clinicopathological characteristics of ITH-low subtypes across all three independent tumor series analyzed. This balance is described and discussed in the manuscript.

Reviewer #3 (Remarks to the Author):

My previous comments have been addressed to my satisfaction.

I reviewed the comments made by Reviewer 2. I think the reviewer raises a good point about the importance of carefully selecting the number k of clusters, which seems to be dataset specific. I think some more discussion about this point might be warranted.

Authors' response: Please refer to our response to Reviewer 2.